# Kinetic drop friction

Xiaomei Li[1], Francisco Bodziony[2], Mariana Yin[2], Holger Marschall[2], Rüdiger Berger ®[1] & Hans-Jürgen Butt ®[1] ✉

Liquid drops sliding on tilted surfaces is an everyday phenomenon and is important for many industrial applications. Still, it is impossible to predict the drop's sliding velocity. To make a step forward in quantitative understanding, we measured the velocity ($U$), contact width ($w$), contact length ($L$), advancing ($\theta_a$), and receding contact angle ($\theta_r$) of liquid drops sliding down inclined flat surfaces made of different materials. We find the friction force acting on sliding drops of polar and non-polar liquids with viscosities ($\eta$) ranging from $10^{-3}$ to 1 Pa · s can empirically be described by $F_f(U) = F_0 + \beta w \eta U$ for a velocity range up to 0.7 ms$^{-1}$. The dimensionless friction coefficient ($\beta$) defined here varies from 20 to 200. It is a material parameter, specific for a liquid/surface combination. While static wetting is fully described by $\theta_a$ and $\theta_r$, for dynamic wetting the friction coefficient is additionally necessary.

When designing surfaces, people commonly use contact angles to characterize surface wettability[1–3]. A low contact angle indicates a high affinity between liquid and solid, and vice versa. In equilibrium, this relationship is expressed by Young's equation[4]. For many applications, it is important how drops slide over surfaces. However, a higher contact angle does not necessarily imply lower friction between a drop and a solid. For example, rose petal and Salvinia leaves have high contact angles but also high lateral adhesion[5–7]. In these cases, the contact angle hysteresis better describes surfaces. Contact angle hysteresis is defined as the difference between the advancing contact angle and the receding contact angle. The static advancing and receding contact angles are measured at the front and rear, respectively, of a sessile drop just before it starts sliding[8]. However, both contact angle and contact angle hysteresis are insufficient to describe the drop dynamics once a drop has started sliding over a surface.

There are many methods to study drop dynamics, such as using de-/inflated drops[9], magnetically controlled oscillated drops[10], direct force measurements with force sensors[11,12], and sliding drops on a tilted surface[13]. When drops slide down tilted surfaces, the external gravitational force driving the motion, $F_g = mg \sin \alpha$, can be adjusted by the tilt angle ($\alpha$), to control droplet velocity in a wide range. Here, $m$ is the mass of the drop, and $g = 9.81$ms$^{-2}$ is the acceleration of gravity[14,15]. Despite many experimental and theoretical studies, it is still impossible to quantitatively predict the forces, which slow down drop motion. We combine these forces, which resist drop motion, in the term "friction force"[16–18]. This friction force is caused by several effects, including hydrodynamic viscous dissipation in the bulk and wedge[19], contact-line friction due to thermal activation of liquid molecules near the contact line[20], pining/de-pinning by inhomogeneity on the surface[21], elastocapillary deformation on soft surfaces[22], surface adaptation[13,23,24], electrostatic retardation induced by slide electrification[25,26], and aerodynamic resistance[27] (Fig. 1). Here, we focus on surfaces, which are flat, smooth, homogenous, rigid, and inert. In this way, we minimize the effects due to pining/de-pinning, elasto-capillary deformation, and adaptation. We chose high-permittivity or conductive substrates to minimize electrostatic retardation. The aerodynamic resistance only becomes substantial for superhydrophobic surfaces, where drops reach velocities higher than 1 ms$^{-1}$. Thus, in our case, only dissipation from viscous and contact lines is expected to be relevant. The questions addressed were: how the drop friction depends on the velocity; which material parameters influence drop friction; How to describe friction forces quantitatively; which dissipation processes contribute how strongly to the friction of sliding drops. The aim is to predict drop sliding velocity quantitatively.

To answer these questions, we recorded drops of 17 liquids with different viscosity ($\eta$) and surface tension ($\gamma$) sliding down 7 different types of planar solid surfaces. We measured drop velocity ($U$), widths ($w$) and lengths ($L$) of their contact area, their advancing ($\theta_a$) and receding contact angles ($\theta_r$). Using the equation of motion, we calculated the friction force ($F_f$). It turned out that friction forces of drops sliding on flat solid surfaces can empirically be described by $F_f = F_0 + \beta w U \eta$. The dimensionless friction coefficient ($\beta$) depends on specific liquid/surface combinations. At least two different channels of

[1]Max Planck Institute for Polymer Research, Ackermannweg 10, 55128 Mainz, Germany. [2]Computational Multiphase Flows, Technische Universität Darmstadt, Alarich-Weiss-Straße 10, 64287 Darmstadt, Germany. ✉e-mail: butt@mpip-mainz.mpg.de

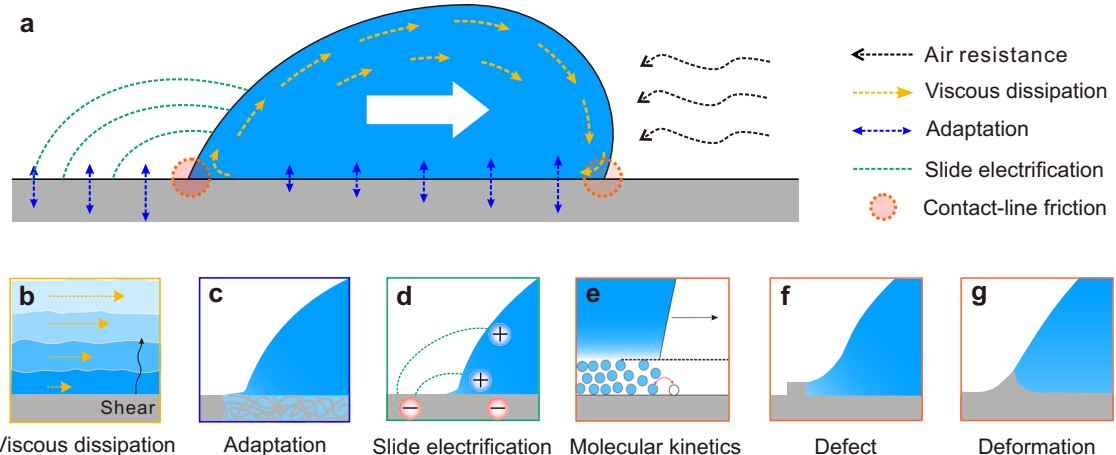

**Fig. 1 | Schematic of all the possible effects contributing to drop friction. a** A sliding drop is slowed down by the effect of air resistance, **b** viscous dissipation, **c** adaptation, **d** slide electrification, and contact-line friction due to **e** molecular kinetics, **f** defects, or **g** sample deformation.

energy dissipation occur: capillary forces caused by contact angle hysteresis and viscous forces caused by shear flow, which are further confirmed by direct numerical simulation. With the empirical equation, a quantitative prediction of drop motion is achieved. Applying the empirical equation facilitates surface design, characterization, and drop manipulation.

## Results

### Empirical description of friction forces

Drops sliding down tilted surfaces typically accelerate and then reach a steady-state velocity. Depending on viscosity and tilt angle, a steady-state is reached after either a short or long slide distance. For example, water drops sliding down Teflon-gold surfaces show a monotonically increasing velocity with sliding time and tilt angle (Fig. 2a). On our observation length of 4 cm, the acceleration phase is not over. Only for very low tilt angles of 10°, water drops reach their steady-state velocity. Water has a viscosity of $\eta = 0.92 \times 10^{-3}$ Pa · s = 0.92 cSt at 25 °C. In contrast, drops of silicone oil with a viscosity of 10 cSt slide down the Teflon-gold surface with steady-state velocities at all tilted angles (Fig. 2b).

We define drop velocity ($U$) as the average velocity of advancing contact-line velocity ($U_a$) and receding contact-line velocity ($U_r$), that is $U = \frac{U_a + U_r}{2}$ (Fig. 2c). For an increasing drop velocity, the drops become longer and narrower. The rear and front velocities are slightly different. The typical difference is, however, less than 8% (Supplementary Fig. 1). The limitation at high velocity is given by two factors. One is the reachable highest tilt angle of the setup of 70°. The second one is pearl formation with tiny satellite droplets behind the primary drop[28]. In this regime, the drop loses its characteristic shape. For this reason, even water drops did not exceed a velocity of ≈0.7 ms⁻¹.

After smoothening measured velocity-versus-time curves, the acceleration ($dU/dt$) of sliding drops was calculated. We extract the friction force $F_f$ on the sliding drop by applying the equation of motion[25]:

$$F_f = mg\sin\alpha - m^*\frac{dU}{dt} \quad (1)$$

Here, $m^*$ is the effective mass of the drop with the consideration of its rolling component. We take $m^*/m = 1.05$, determined by diffuse-interface phase-field method[25]. We neglect the effect that $m^*/m$ slightly changes with velocity because the error from varying $m^*/m$ is lower than the variation of velocity observed from sample to sample (Supplementary Fig. 2). We also assume that the drop shape has reached its steady-state at every velocity. This assumption is not

entirely true, since the drop is accelerating and the real drop shape slightly lags behind. In addition, the damped drop oscillations may cause a deviation from the steady-state drop shape.

We measured drops of 17 different liquids with viscosities ranging over three orders of magnitude (Table 1). Seven different surfaces were studied: (1) naturally oxidized bare silicon wafers (Si wafer), (2) indium tin oxide (ITO) coated glass, (3) 1H, 1H, 2H, 2H-perfluorooctadecyltrichlorosilane (PFOTS) on a silicon wafer, (4) polydimethylsiloxane (PDMS) brushes on silicon wafers, (5) 35 nm polystyrene (PS) on gold, (6) perfluorodecanethiol monolayer (thiols) on gold and (7) 60 nm Teflon on gold. The topography of the surfaces imaged by scanning force microscopy (SFM) shows a surface roughness between 0.1 and 2.5 nm (Supplementary Fig. 3). The sliding details about drop shape, drop velocity, contact length, contact width, kinetic advancing contact angle, kinetic receding contact angles, and forces are summarized in Supplementary Figs. 4–25.

For all liquid/surface combinations, the friction force increases with drop velocity. To find a universal empirical equation that describes the velocity-dependent friction force, we plot friction force per unit width versus the velocity multiplied by viscosity. It is equivalent to friction force per unit width and surface tension versus capillary number, $Ca = U\eta/\gamma$ (Fig. 2d). Normalization of the friction force by the contact width is reasonable because the capillary force is proportional to the width of the contact area of the drop (see below). In addition, scaling with viscosity allows the comparison of different liquids. All graphs in Figs. 2d and 3 exhibit a linear dependence to a large degree. The friction force for all the liquid/surface combinations is accordingly fitted with the equation:

$$F_f = F_0 + \beta wU\eta \quad (2)$$

Here, $F_0$ is the friction force extrapolated to $U = 0$. We call the dimensionless term $\beta$ friction coefficient. The terminology of friction coefficient here is different from Bocquet and Barrat's[29], de Ruijter's[30], or McHale's definition[17]. Bocquet and Barrat defined a phenomenological friction coefficient with the symbol $\lambda$ as the ratio of friction force to velocity by the hydrodynamic approach for the liquid/solid boundary with a unit of Nsm⁻¹²⁹. de Ruijter defined a friction coefficient per unit length of the contact line with the symbol $\zeta_0$ based on the molecular kinetic theory for drop spreading in the unit of Pa · s[30]. Similar to the solid/solid system, McHale defined the ratio of drop friction to its normal adhesion as the friction coefficient represented by $\mu$[17]. With the definition by Eq. (2), the friction coefficient depends on the specified liquid/solid surface combination (Fig. 2e and Table 1). It

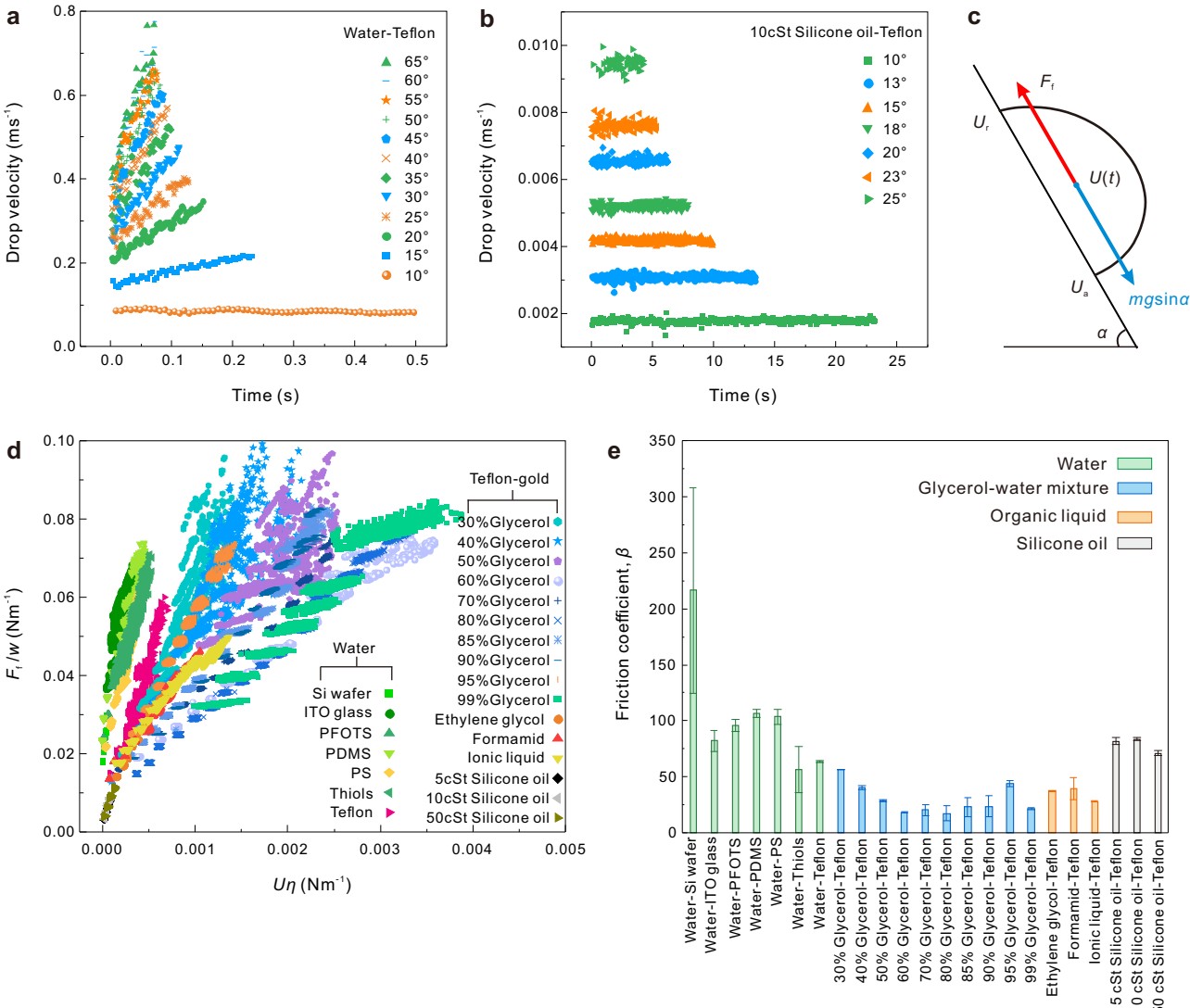

**Fig. 2 | Calculation and description of friction force. a** Drop velocity-versus-time for 30 µl water drops on Teflon-gold surfaces measured at different tilt angles. **b** Drop velocity-versus-time for 10 µl 10 cSt silicone oil drops on Teflon-gold surfaces measured at different tilt angles. **c** Schematic of forces acting on a sliding drop. In the steady-state, $F_f = mg \sin \alpha$. In the acceleration phase, $m^* \frac{dU}{dt}$ was taken into account. **d** Friction forces per unit width ($F_f/w$) versus velocity multiplied by its viscous ($U\eta$) for 23 liquid/surface systems. The details of most curves are provided in Fig. 3. **e** The friction coefficient ($\beta$) of 23 liquid/surface systems. The friction coefficient here is defined in Eq. (2), different from the ones in literature[17,29,30]. Error bars present the standard deviation of $\beta$ from at least two measurements.

indicates how the friction force increases with velocity during drop sliding and can act as an material parameter characterizing drop sliding for a certain liquid/solid combination.

## Contribution of capillary and viscous force to drop friction

The friction force acting on a static sessile drop is given by the integral of the lateral surface tensional forces acting around the contact line (Fig. 4a)[31–34]:

$$F_c = 2\gamma \int_0^\pi \xi \cos \theta \cos \varphi \, d\varphi \tag{3}$$

Here, $\xi$ is the radius describing the position of the contact line, $\varphi$ is the azimuthal angle, $\theta$ is the contact angle. After integration, one obtains:

$$F_c = w\gamma k \left( \cos \theta_r - \cos \theta_a \right) \tag{4}$$

in which, $k$ is a geometric factor, whose precise value depends on the shape of the drop[35]. Equation (4) is often referred as the Furmidge-Kawasaki equation[24,36,37]. Here we call it capillary force $F_c$.

Therefore, the static friction force per unit width $F_0/w$ on a sessile drop is calculated by $k\gamma(\cos \theta_{as} - \cos \theta_{rs})$. We obtain $F_0/w$ with Eq. (2) by extrapolating measured kinetic friction forces to $U \to 0$. Using the static advancing ($\theta_{as}$) and receding contact angles ($\theta_{rs}$) measured by the in-/deflated drop ("Method" section and Table 1), we calculate the static $k$-factor ($k_s$):

$$k_s = \frac{F_0/w}{\gamma(\cos \theta_{as} - \cos \theta_{rs})} \tag{5}$$

We find that $k_s$ lies in the range of 0.5 to 1.5 for all the liquid/surface combinations within the error of the measurements (Fig. 4c). The average $k$-factor of all the liquid/surface systems is $0.88 \pm 0.2$ for the onset of sliding. This is consistent with ElSherbini and Jacobi's calculation ($k = 24/\pi^3 = 0.774$)[33] and Extrand's experimental results

**Table 1 | Properties of all the liquid/surface combinations**

| System number | Liquid-Surface | $m^c$ mg | $\gamma$ mNm$^{-1}$ | $\eta^d$ mPa · s | $\theta_{as}^{e}$ ° | $\theta_{rs}^{e}$ ° | $\beta^f$ | $F_0/w^g$ mNm$^{-1}$ | $k_s^h$ |
|---|---|---|---|---|---|---|---|---|---|
| 1 | Water-Si wafer | 30 | 72 | 0.92 | 65 | 35 | 216 ± 92 | 18.0 ± 2 | 0.63 ± 0.022 |
| 2 | Water-ITO glass | 30 | 72 | 0.92 | 111 | 83 | 82 ± 9 | 34.3 ± 1 | 0.99 ± 0.007 |
| 3 | Water-PFOTS | 30 | 72 | 0.92 | 116 | 86 | 96 ± 5 | 24.7 ± 1 | 0.95 ± 0.022 |
| 4 | Water-PDMS | 30 | 72 | 0.92 | 108 | 87 | 107 ± 4 | 25.6 ± 1 | 0.7 ± 0.013 |
| 5 | Water-PS | 30 | 72 | 0.92 | 95 | 78 | 104 ± 7 | 19.1 ± 1 | 0.9 ± 0.012 |
| 6 | Water-Thiols | 30 | 72 | 0.92 | 120 | 92 | 56 ± 21 | 33.3 ± 5 | 0.99 ± 0.069 |
| 7 | Water-Teflon | 30 | 72 | 0.92 | 122 | 110 | 63 ± 1 | 8.0 ± 1 | 0.59 ± 0.051 |
| 8 | 30% Glycerol-Teflon | 30 | 69 | 2.5 | 112 | 102 | 56 ± 1 | 6.6 ± 1 | 0.57 ± 0.022 |
| 9 | 40% Glycerol-Teflon | 30 | 69 | 3.8 | 111 | 101 | 41 ± 2 | 10.0 ± 2 | 0.86 ± 0.050 |
| 10 | 50% Glycerol-Teflon | 30 | 68 | 6.9 | 109 | 100 | 29 ± 1 | 13.3 ± 1 | 0.97 ± 0.027 |
| 11 | 60% Glycerol-Teflon | 30 | 67 | 13.6 | 113 | 101 | 18 ± 1 | 13.2 ± 1 | 0.99 ± 0.011 |
| 12 | 70% Glycerol-Teflon | 30 | 66 | 27.1 | 112 | 101 | 20 ± 5 | 13.1 ± 1 | 1.08 ± 0.053 |
| 13 | 80% Glycerol-Teflon | 30 | 66 | 75.9 | 112 | 102 | 17 ± 7 | 10.8 ± 5 | 0.98 ± 0.190 |
| 14 | 85% Glycerol-Teflon | 30 | 65 | 93 | 111 | 100 | 23 ± 9 | 12.2 ± 3 | 1.01 ± 0.084 |
| 15 | 90% Glycerol-Teflon | 30 | 65 | 192 | 111 | 101 | 23 ± 10 | 8.9 ± 2 | 0.81 ± 0.065 |
| 16 | 95% Glycerol-Teflon | 30 | 65 | 265 | 109 | 99 | 44 ± 3 | 10.5 ± 2 | 0.87 ± 0.040 |
| 17 | 99% Glycerol-Teflon$^a$ | 30 | 64 | 943 | 111 | 98 | 22 ± 1 | 6.1 ± 1 | 0.43 ± 0.048 |
| 18 | Ethylene glycol-Teflon | 21 | 48 | 16 | 98 | 88 | 38 ± 1 | 9.6 ± 1 | 1.15 ± 0.015 |
| 19 | Formamide-Teflon | 32 | 58 | 4.6 | 105 | 94 | 40 ± 10 | 9.0 ± 4 | 0.82 ± 0.140 |
| 20 | Ionic liquid$^b$-Teflon | 30 | 51 | 22 | 101 | 90 | 28 ± 1 | 14.3 ± 1 | 1.46 ± 0.079 |
| 21 | 5 cSt Silicone oil-Teflon | 10 | 21 | 5 | 55 | 45 | 82 ± 3 | 1.9 ± 1 | 0.68 ± 0.012 |
| 22 | 10 cSt Silicone oil-Teflon | 11 | 21 | 10 | 56 | 49 | 84 ± 1 | 2.1 ± 1 | 1.01 ± 0.007 |
| 23 | 50 cSt Silicone oil-Teflon | 12 | 21 | 50 | 58 | 47 | 71 ± 2 | 2.7 ± 1 | 0.8 ± 0.023 |

$m, \gamma, and\ \eta$ are the drop mass, surface tension, and viscosity. $\theta_{as}$ and $\theta_{rs}$ are the static advancing and receding contact angles. $\beta$ is the friction coefficient. $\frac{F_0}{w}$ is the kinetic friction force at zero velocity per unit width. $k_s$ is the static $k$-factor.
$^a$99% glycerol was purchased commercially while 30–95% glycerol-water mixtures were made with 99% glycerol and distilled water.
$^b$The ionic liquid is 1-ethyl-3-methyl-imidazolium-thiocyanate.
$^c$The uncertainty of the drop-mass measurement is typically ±3 mg.
$^d$The uncertainty of the viscosity measurement is around 15%.
$^e$The uncertainty of the static contact angle measurement is roughly ±3°.
$^f$" ± " presents the standard deviation of $\beta$ from at least two measurements. The friction coefficient is defined in Eq. (2), different from the ones in literature[17,29,30].
$^g$" ± " presents the standard deviation of $F_0/w$ from at least two measurements.
$^h$" ± " presents Gaussian error propagation of the standard deviation of $\theta_{as/rs}$ and $F_0/w$.

($k = 4/\pi$ when using drop radius instead of drop width)[38,39]. For a hypothetical two-dimensional drop (Fig. 4b), the two parallel sides which are parallel to the external force do not contribute to the retentive force, resulting in $k = 1$[36,37,40–42]. In reality, $k$ depends on the shape of the contact line and on how the actual contact angle varies along the contact line[33,38,39,43]. Consequently, for sessile drops, a $k$-factor <1 is expected. As an alternative, Tadmor proposed to use the length prefactor $L_p = A/h$ rather than the width of the drop in Eq. (4) to calculate the friction force of a sliding drop[24]. Here, $A$ is the contact area of the drop, and $h$ is its height. Since the contact area cannot be measured by our present setup directly, it would be interesting to check the possibility in the future. We tried to estimate the correlation between $L_p$ and the $k_s$. It does however, not lead to a significant correlation (Supplementary Fig. 26).

Can one apply Eq. (4) also in the kinetic regime? In the kinetic regime, drops elongate with increasing velocity (Fig. 4d and Supplementary Figs. 4–25). As a result, the contact length ($L$) increased while contact width ($w$) decreased (Fig. 4e and Supplementary Figs. 4–25), leading to an increasing aspect ratio ($L/w$) with increasing velocity. The velocity-dependent aspect ratio can be fitted by a 2nd-order polynomial (Fig. 4f and Supplementary Figs. 4–25). Knowing the relationship between aspect ratio and drop velocity, and by measuring drop velocity and contact length from the side view, we determined the contact width. At the same time, the kinetic advancing contact angles increase while the kinetic receding contact angles decrease with drop velocity

(Fig. 4g and Supplementary Figs. 4–25). By inserting $\theta_r(U)$, $\theta_a(U)$, and $w(U)$ into Eq. (4), the capillary force ($F_c$) is calculated. For simplicity, we assume $k = 1$ and is independent of velocity. The capillary forces as given by Eq. (4) are a good estimate for measured friction forces for all tested liquids and surfaces (Figs. 3 and 4h and Supplementary Figs. 4–25). The absolute error is around 5–25% with $k = 1$ (Fig. 4i). Thus, Eq. (4) is a good description of the friction force and holds even in the kinetic regime by taking an appropriate $k$-factor.

The above-mentioned proportionality of the friction force to velocity and viscosity indicates that hydrodynamic dissipation plays a substantial role. For this reason, it is not a priori clear why Eq. (4) is able to describe the kinetic friction force of sliding drops. A possible explanation is that viscous dissipating processes influence the contact angles, which enter Eq. (4). With our setup, we measured the contact angles optically on a length scale of 10–100 μm. Therefore, energy dissipation occurring closer than 100 μm to the contact line is taken into account. At this scale, wedge viscous dissipation should show up as a change in contact angle and via Eq. (4) in friction (Fig. 4j).

Assuming the contact-line friction and wedge viscous dissipation enter via the kinetic advancing and receding contact angles leaves us with a total friction force:

$$F_f = w\gamma k \left( \cos\theta_r - \cos\theta_a \right) + F_{v-bulk} \qquad (6)$$

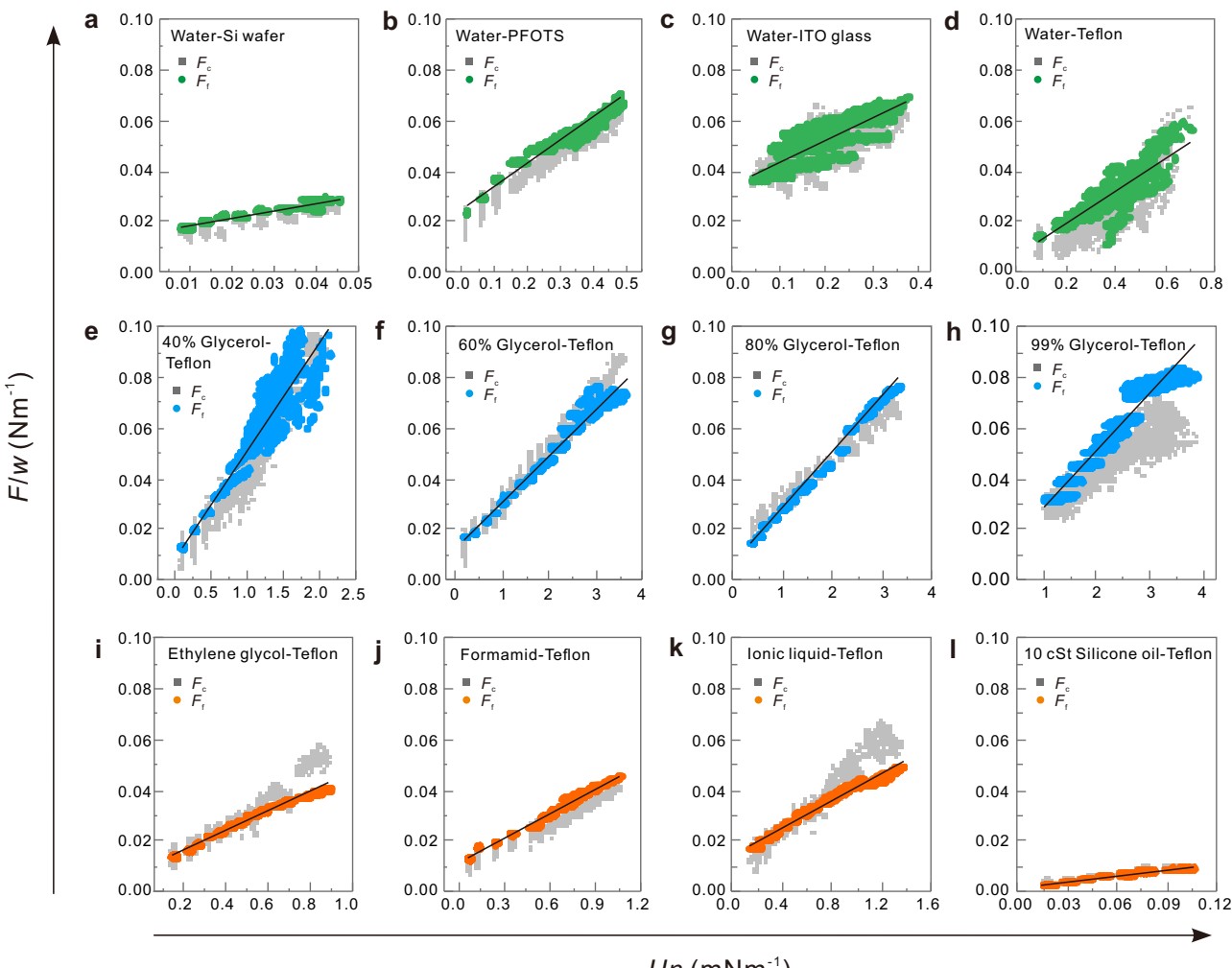

**Fig. 3 | Linear relationship between friction force and drop velocity. a–l** Friction force ($F_f$) and capillary force ($F_c$) per unit width ($w$) of the drop-versus-velocity multiplied by viscosity for different liquids on different surfaces. Black curves are linear fits with Eq. (2). The corresponding fitting parameters of $\beta$ and $F_0/w$ are shown in Table 1. Capillary forces were calculated with Eq. (4) with measured $\theta_a(U)$, $\theta_r(U)$, and $k=1$. In some cases like 60–99% glycerol-water mixture or silicone oil, the drop reached its steady-state velocity within the first mm of sliding so that $dU/dt{\approx}0$. As a result, the graph $F_f/w$-vs.-$U\eta$ looks non-continuous. In contrast, for low-viscosity liquids such as water, drops accelerated within the whole recorded slide length, leading to continuous $F_f/w$-vs.-$U\eta$ graphs.

Here, $F_{v-bulk}$ is the bulk viscous force. The good agreement between the forces calculated with the Furmidge-Kawasaki Eq. (4) and measured friction forces is thus most likely the effect of two compensating errors. On one hand, by setting $k=1$, we may overestimate the capillary contribution. On the other hand, we neglect bulk viscous forces, leading to an underestimation of the friction force.

**Forces of sliding drop in the simulation**

Separating viscous dissipation from the wedge and bulk is a challenge as the transition between both is gradual[19]. To find out where viscous energy is dissipated inside a sliding drop, we carried out direct numerical simulations. The force caused by viscous dissipation can be accounted for by integrating the hydrodynamic shear stress (viscous force density in Nm⁻²) over the contact area of the drop. The simulations show that viscous dissipation increases close to the three-phase contact line (Fig. 5a). Thus, we define the "wedge" region with a height of mesh height (37 μm) and a width (100–350 μm) equal to the double distance from the contact line to the peak of viscous dissipation (Fig. 5b). With this definition, the ratio between the wedge and bulk viscous dissipation is around 1/1 for water drops on PS-gold surfaces (Fig. 5c, green and blue

triangles). For drops of 85% glycerol-water mixture on Teflon-gold surfaces, the wedge-to-bulk dissipation is 7/3 (Supplementary Fig. 28). Both bulk and wedge dissipation forces increase roughly linearly with velocity (Fig. 5c, green and blue triangles). Due to numerical errors inherent to the phase-field approach, the linearly increasing viscous forces deviate slightly at high velocity. Most importantly, bulk viscous forces in both systems are the lowest dissipated force and take up less than 20% of the friction force. Such a low contribution of bulk viscous force is in line with the deviation of capillary force from friction force (Fig. 4i) and confirms our hypothesis that bulk viscous force is low.

The capillary force in the simulation was calculated by integrating the capillary stress, which is a function of the surface tension, over the contact area (Eq. (10)). The capillary force is almost constant for the 85% glycerol-water mixture (Supplementary Fig. 28, star). It increases linearly for pure water (Fig. 5c, star). The increasing capillary force for water can be attributed to the elongation of the drop with increasing contact area at high velocities (>0.1 ms⁻¹). The other reason could be that the shape of the interfacial profile degrades with fast dynamics at high velocities (details in the "Experimental" section). This degradation could be reduced

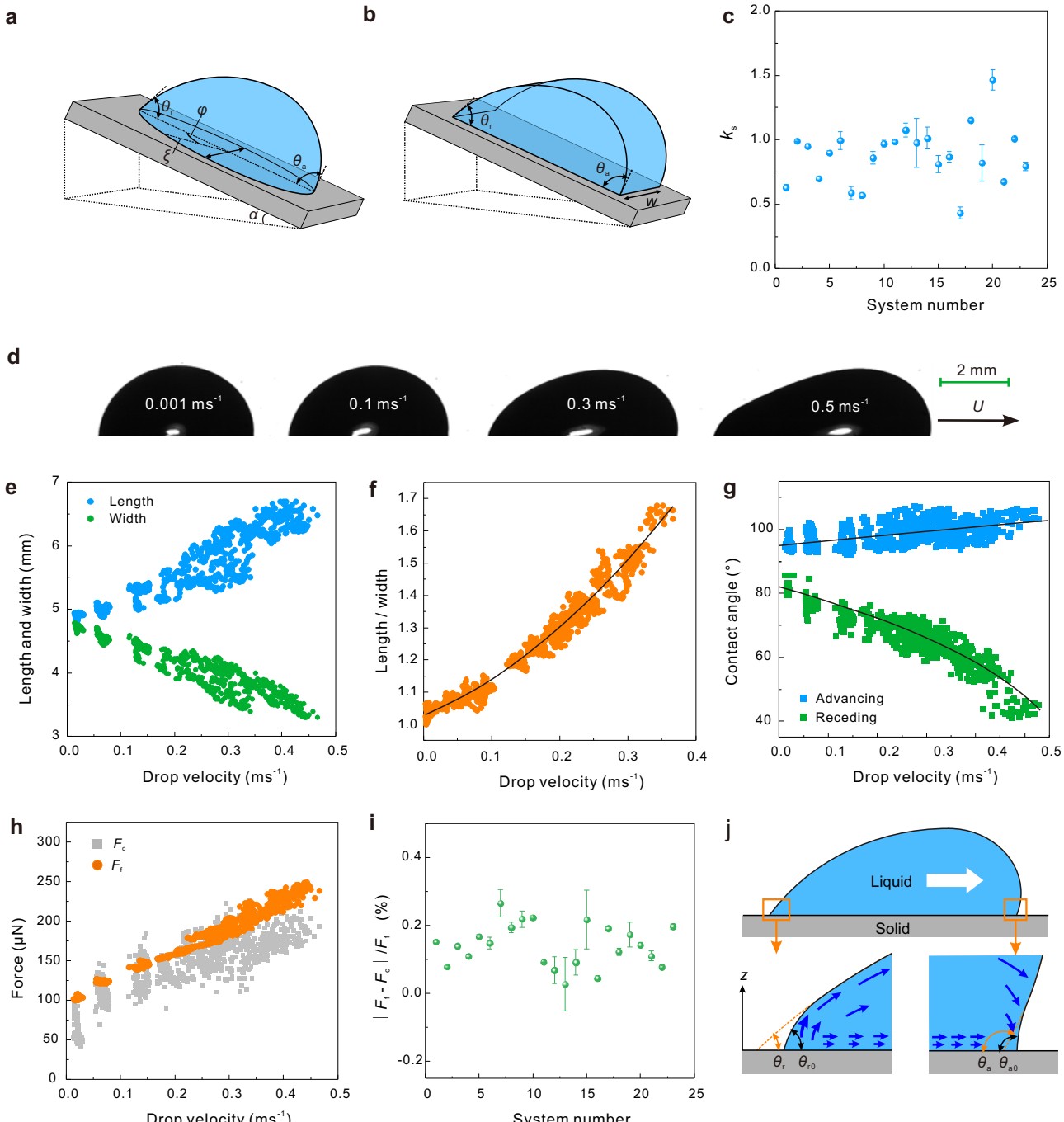

**Fig. 4 | Kinetic capillary force and its related parameters. a** Schematic of a drop on a tilted plate and its two-dimensional counterpart (**b**). **c** Static $k$-factor ($k_s$) versus system number derived for the onset of sliding. $k_s$ was calculated by Eq. (5). Error bars correspond to Gaussian error propagation of the standard deviation of $\theta_{as/rs}$ and $F_0/w$. **d** Side-view images of sliding water drops on the PS-gold surfaces at different velocities. **e** Velocity-dependent length and width of the liquid-solid contact area, and **f** length-to-width for 30 μl water drops on PS-gold surfaces. The black curve results from a fit of $\frac{L}{w} = 2.049U^2 + 1.072U + 1.018$. **g** Velocity-dependent kinetic advancing and receding contact angles for 30 μl water drops on PS-gold surfaces. The black curves are fitted by Eq. (7). The fitting parameters ($\theta_{a0}$, $\theta_{r0}$, $\frac{l}{l_m}$) are summarized in Supplementary Table 1. **h** Friction force and capillary force versus drop velocity for water drops sliding on PS-gold surfaces. **i** The percentage of the absolute difference between $F_c$ and $F_f$ in $F_f$ ($|F_f - F_c|/F_f$) for 23 liquid/surface combinations. Error bars indicate the standard deviation after averaging the percentages at different velocities. **j** Schematic of the shape of the drop close to the receding and advancing contact line. $\theta_{a0/r0}$ is the microscopic advancing or receding contact angle.

by implementing an interfacial relaxation method like the one proposed in ref. 44. In contrast, the velocity for the 85% glycerol-water mixture is so low that the shape of the drops did not change much. We further compare the capillary force in the experiment with the sum of simulated wedge viscous force and simulated capillary forces (Fig. 5c and Supplementary Fig. 28, circle). They

match well, further confirming our hypothesis that capillary force includes the wedge viscous dissipation.

## Discussion

Viscous dissipation brought by shear flow near the contact line has been modeled by Cox, Voinov, and others using continuum

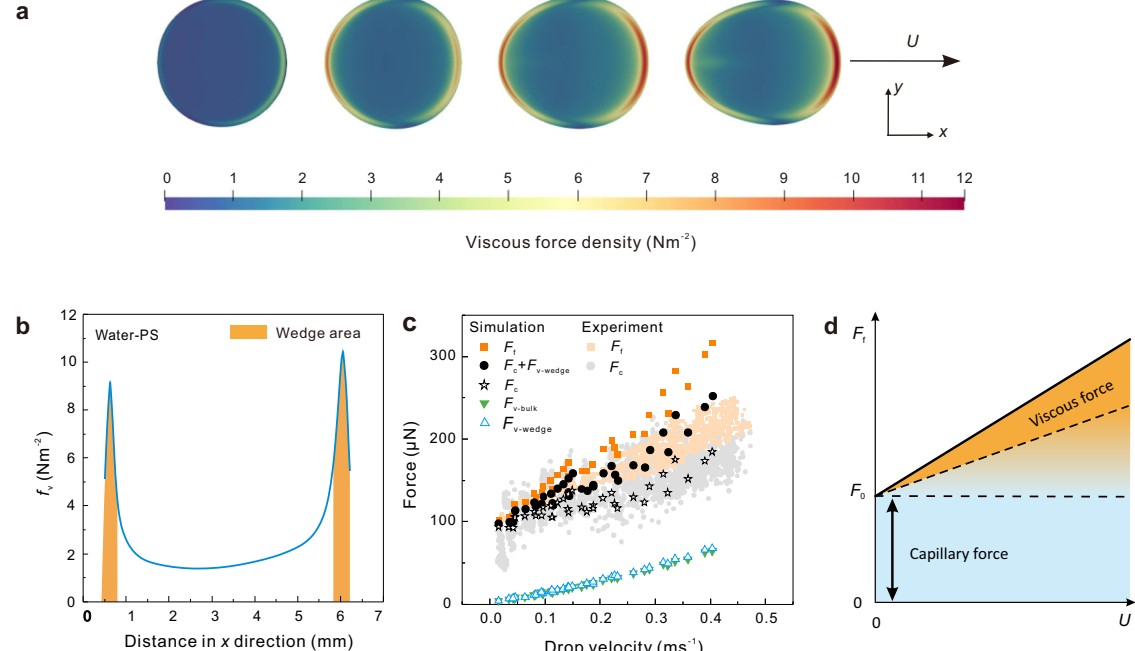

**Fig. 5 | Simulation of a sliding drop by direct numerical simulations. a** The distribution of viscous force density ($f_v$) at the solid-liquid interface of a water drop on a 50° tilted PS-gold surface at velocities at 0.06, 0.12, 0.18, and 0.25  ms⁻¹ from left to right. **b** The definition of wedge region which is based on the viscous force density along $x$ direction in (**a**). **c** Comparison of simulated force with the experimental force for water drops on PS-gold surface. $F_f$, $F_c$, $F_{v-wedge}$, and $F_{v-bulk}$ are the friction force, capillary force, wedge viscous force, and bulk viscous force correspondingly. **d** Schematic of the friction force and its origin as a function of the velocity of sliding drops.

hydrodynamics[45–47]. To check the validity of their model, we fit the kinetic contact angles with the Cox−Voinov theory[45,46]:

$$\theta_{a/r}(U) = \left( \theta_{a0/r0}^3 \pm 9Ca \ln \frac{l}{l_m} \right)^{1/3} \quad (7)$$

Here, $\theta_{a0/r0}$ is the microscopic advancing or receding contact angle. $l$ is a macroscopic cut-off length related to the drop size. $l_m$ is a microscopic cut-off length, usually of molecular scale, below which slip is allowed. With $l \approx 1$mm and $l_m \approx 1$nm, a ratio of the order of $l/l_m \approx 10^6$ is expected. The fitting of the receding contact angle for water leads to 10 orders of magnitude higher values for $l/l_m$ (Fig. 4g, Supplementary Figs. 4−S25c and Supplementary Table 1). We conclude that the Cox−Voinov theory is insufficient to describe the velocity-dependent contact angle quantitatively for water.

To further check the Cox−Voinov theory qualitatively, we calculate the friction force versus capillary number $Ca$ assuming that the change in contact angles is given by Eq. (7). It turns out that the hydrodynamic contribution to drop friction can be well approximated as being linear up to $Ca$ of the order of 0.01−0.02 (Supplementary Fig. 27a). McHale et al. already considered the linear part of $F_c$-vs.-$Ca$ graphs by inserting Eq. (7) into Eq. (4) and taking the first element of a Taylor series[17]. In analogy with their approach, by setting friction force ($F_f$) in Eq. (2) equal to the capillary force in Eq. (4), we obtain:

$$\beta' = 3k \ln \frac{l}{l_m} \cdot \left[ \frac{\sin \theta_{a0}}{\theta_{a0}^2} + \frac{\sin \theta_{r0}}{\theta_{0r0}^2} \right] \quad (8)$$

The superscript in $\beta'$ is to distinguish from the empirical $\beta$ listed in Table 1. When estimating the friction coefficient with typical parameters for water-glycerol mixtures ($\theta_{a0} = \theta_{r0} = 100°$, $l/l_m = 10^6$, $k = 1$), we get $\beta' \approx 22$. For silicon oils ($\theta_{a0} = \theta_{r0} = 50°$, $l/l_m = 10^6$, $k = 1$), we get $\beta' \approx 70$. The order of magnitude agrees with experimental values

(Table 1). But there is no significant correlation between $\beta$ and the contact angle due to the scattering data points (Supplementary Fig. 27b).

These above results indicate that viscous dissipation as modeled by Cox and Voinov is insufficient to describe drop friction. It might be due to other effects such as contact-line friction[20,48,49] or adaptation[23] contributing substantially. When the contribution from other effects increases substantially, the linear relationship between drop friction and velocity might change. For example, Keiser et al.[50] and Smith et al.[51] reported a nonlinear relationship between drop friction and velocity on liquid-infused surfaces due to the velocity-dependent change in the shape of the meniscus.

In summary, at least two different channels of energy dissipation occur: capillary forces caused by contact angle hysteresis and viscous forces caused by shear flow. At low velocity, the capillary force dominates (Fig. 5d, blue region). It is given by $F_f(U \to 0) = F_0 \approx 0.88 w \gamma (\cos \theta_{rs} - \cos \theta_{as})$. With increasing velocity, the linearly increasing part of the friction force, $\beta w U \eta$, contributes more and more (Fig. 5d, orange region). This increase can be explained by increasing wedge and bulk viscous dissipation. We can, however, not exclude the contribution from other effects, such as overcoming local energy barriers, adaptation, or remaining electrostatic retardation. Water is an exception. Since the velocity-dependence of the contact angles cannot be fitted with Cox−Voinov theory, we conclude that viscous dissipation cannot fully account for drop friction and other effects contribute substantially.

## Methods
### Liquids
As liquids, we used distilled water (<1 μScm⁻¹; Gibco, Thermo Fisher Scientific), glycerol (99%, AppliChem), water-glycerol mixtures, ethylene glycol (≥99%, VWR Chemical), formamide (99.5%, AppliChem), the ionic liquid 1-ethyl-3-methyl-imidazolium-thiocyanate (≥95%, Sigma-Aldrich), and silicone oil (Sigma-Aldrich).

## Preparation of surfaces

To minimize electrostatic effects, all surfaces were prepared on substrates with high dielectric permittivity. We analyzed seven types of planar surfaces. (1) Si wafer: Si wafer with a native oxide layer of $1.6 \pm 0.3$ nm as measured by ellipsometry, resistivity $<0.005\,\Omega\,cm^{-1}$, and thickness of $525 \pm 25\,\mu m$ (SiMat, Germany). After being cut into $25 \times 100$ mm$^2$, large pieces, they were cleaned by ultrasonication in ethanol (absolute, VWR Chemical). Then they were dried by nitrogen blowing; (2) ITO glass: ITO glass ($24 \times 60 \times 0.175$ mm$^3$) is from Präzisions Glas & Optik (Germany) with a resistivity of $20 \pm 5$ Ohm. We used the ITO glass without further processing; (3) PFOTS-Si surfaces: the PFOTS coating on the Si wafer was prepared by chemical vapor deposition. The clean Si wafers were activated by a 100% $O_2$ plasma for 10 min. Then we put the Si wafers into a vacuum desiccator containing a tiny glass bottle with 0.5 ml 1H, 1H, 2H, 2H-perfluorooctadecyltrichlorosilane (97%, Sigma-Aldrich). The desiccator was evacuated to <100 mbar. After 30 min, the samples were taken out and cleaned by rinsing with ethanol (absolute, VWR Chemical) to remove unbound silanes; (4) PDMS-Si surfaces: the polydimethylsiloxane (PDMS) brushes coatings with a thickness of around 5 nm were prepared by the "grafting to" method using PDMS (molecular weight, 6 kg mol$^{-1}$; Alfa Aesar) as described in ref. [52]. A few drops of PDMS were deposited on a clean Si wafer. Then samples were stored at 22–23 °C and 30–60% relative humidity for 24–48 h after the PDMS drops spread and covered the substrates. Before use, they were cleaned by ultrasonication in toluene (99.8%, Fisher Chemical), ethanol (absolute, VWR Chemical), and distilled water for 10 min each. The "graft to" method used here to prepare PDMS brushes is different from the "graft from" method to prepare PDMS brushes[53] or PDFMS brushes[54], which might lead to different contact angle hysteresis; (5) PS-gold surfaces: gold substrates with 5 nm chromium (Cr) and 35 nm gold (Au) on the glass slide were prepared by sputter coating. After sputter coating (BalTec MED 020), the gold substrates were used immediately without further cleaning. Thirty-five nm polystyrene (PS) coatings on gold substrates were prepared by dip-coating at pulling speed of 90 mm min$^{-1}$ from a solution of 1 wt% PS (molecular weight: 192 kg mol$^{-1}$, $\varepsilon = 2.6$; Sigma-Aldrich) in toluene (99.8%, Fisher Chemical). Before use, the PS samples were annealed in the oven at 120 °C under vacuum for 24 h; (6) Thiols-gold surfaces were prepared by immersing fresh gold substrates in 1 mM 1H, 1H, 2H, 2H-perfluorodecanethiol (97%, Sigma-Aldrich)/ethanol (absolute, VWR Chemical) solution for 24 h. Then the surfaces were taken out and rinsed with fresh ethanol (absolute, VWR Chemical) to remove unbound thiols; (7) Teflon-gold surfaces: 60 nm Teflon coatings on gold substrates were prepared by dip-coating at pulling speed of 10 mm min$^{-1}$ from a solution of 1 wt% Teflon AF 1600 ($\varepsilon = 1.9$; Sigma-Aldrich) in FC-75 (97%, Fisher Scientific). Before use, the Teflon samples were annealed in the oven at 160 °C under vacuum for 24 h.

## Measurement of surface roughness

Surface roughness was determined by scanning force microscopy in tapping mode (Dimension Icon, Bruker) on an area of $0.5 \times 0.5\,\mu m^2$ (Supplementary Fig. 3). The cantilever had a nominal resonance frequency of 300 kHz and a spring constant of 26 Nm$^{-1}$ (160AC-NA, OPUS). The errors of root-mean-square (RMS) roughness are from the deviation of three measurements on different positions and different samples.

## Measurement of viscosity

The viscosity of the glycerol-water mixture was measured by a rolling ball viscometer LOVIS 2000 M (Anton Paar) with 600 µl solution at 25 °C.

## Measurement of static advancing and receding contact angles

Method of in-/deflated sessile water droplets was used to quantify the "static" advancing and receding contact angles ($\theta_{as/rs}$) by OCA 35, DataPhysics Instruments. First, an 8 µl liquid drop was deposited on the tested surfaces. Then 16 µl liquid was pumped into then pumped out of the drop with a flow rate of 1 µls$^{-1}$ by a Hamilton syringe with a hydrophobic needle. Without pausing, the procedure was carried out three times. The inflation and deflation of drop were recorded from the side. By elliptical fitting to the drop contour, $\theta_{rs}$, and $\theta_{as}$ were determined.

## Measurement of sliding drop

The velocity and the kinetic contact angles, $\theta_r(U)$ and $\theta_a(U)$ were measured by a home-built tilted plate setup[13,25]. The drops were placed automatically on the tilted surfaces from a grounded syringe needle with 1.5 mm outer diameter connected to a peristaltic pump (MINI-PULS 3, Gilson). Different liquids had slightly different drop sizes (Table 1) because the drop volume depends on the surface tension and density of the liquid. The height between the syringe needle and the surfaces was ≈5 mm, just enough to release the drop. Before starting to slide, the drops were neutralized by a grounded electrode. A high-speed camera (FASTCAM Mini UX100 (Photron) from the side recorded the drop sliding after the drops had detached from the grounded electrode. The lens (TitanTL telecentric lens, ×0.268, C-mount, Edmund Optics) had a resolution of ≈37 µm per pixel. Side view videos of the sliding drops were analyzed by an adapted open drop-shape analysis code from MATLAB (DSAfM) version 9.5.0.944444 (R2018b). The MATLAB code was originally developed by Andersen and Taboryski, and the details as well as the code can be found in ref. [55]. The kinetic advancing and receding contact angles were determined by applying a polynomial fit to two-semi ellipse drop counters, which were divided in the middle of drops. The drop velocity was an average from the rear, $U_r$, and front, $U_a$, contact line velocity. All measurements were conducted at a temperature of $20 \pm 1$ °C and a humidity of 15–30%.

## Direct numerical simulations

The simulations were based on a diffuse interface phase-field method. In this method, an initial hemispherical drop with a radius of $a = 2.5$ mm is placed on a $0.025 \times 0.010$ m$^2$ rectangular smooth inclined wall. The contact angle field of the numerical drop is computed directly from the wetting boundary condition using static contact angle as an input. We used the adaptive mesh refinement technique with a mesh width of around 37 µm and a total mesh cell number of about 400,000. We have tried larger cell numbers of 700,000, but the influence of mesh density on numerical accuracy was negligible. The details about the schemes and solver in the simulation are referred to ref. [56]. To calculate the velocity and acceleration, the drop's barycentre positions were tracked for various inclination angles. Based on the velocity gradient, $\nabla U$, the viscous stress tensor, $\tau = \mu(\nabla U + \nabla U^T)$, and the viscous force, $F_v$, in a contact area domain $\Omega$ was calculated:

$$F_v = \int f_v d\Omega = \int \tau : \nabla U \mathrm{d}\Omega \qquad (9)$$

$f_v$ is the viscous density per unit area in Nm$^{-2}$. The capillary force was computed by integrating the surface tension stress in the contact area domain $\Omega$ with:

$$F_c = \int \sigma \epsilon \partial_n C \nabla C \mathrm{d}\Omega \qquad (10)$$

$\sigma$ relates to the surface tension $\gamma$ by $\sigma = \frac{3}{2\sqrt{2}}\gamma$, $\epsilon$ is the capillary width indicating the thickness of the diffuse interface. $C$ is the phase-field order parameter defined by volume fraction. $C = \pm 1$ represents a pure phase, while $C(x) \in (-1, 1)$ indicates a mixing phase. In one

dimension, when $C(x)$ follows:

$$C(x) = \tanh\left(\frac{x}{\sqrt{2}\epsilon}\right) \qquad (11)$$

the drop interface has an equilibrium profile. We assume our numerical drop has an equilibrium profile to calculate the capillary force, which is not entirely accurate because the interface profile gets degraded with $C(x)$ deviating from Eq. (11) at high velocity ($>0.1\,\mathrm{ms}^{-1}$).

## Data availability
Source datasets for all figures in both the main test and the Supplementary Information are available under the accession code (https://doi.org/10.6084/m9.figshare.23540259). Raw videos that support the dataset are available from the corresponding authors upon request owing to the file size of all videos.

## Code availability
The MATLAB code used for image processing in this work is originally developed by N. K. Andersen and R. Taboryski and adapted by S. Silge. The code is available under the accession code (https://doi.org/10.6084/m9.figshare.23540259).

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

## Acknowledgements

Computations for this research were conducted on the Lichtenberg high-performance computer of the Technical University of Darmstadt. This project has received funding from (1) the Priority Program 2171 "Dynamic wetting of flexible, adaptive and switchable surfaces" (grant no. BU 1556/36 and BE 3286/6-1: X.L., H.-J.B., R.B.); (2) the European Research Council (ERC) under the European Union's Horizon 2020 research and innovation program (grant agreement no. 883631) (H.-J.B., X.L.); (3) the German Research Society via the CRC 1194 (project ID 265191195) "Interaction between transport and wetting processes", project B07 (F.B., H.M.) and C07 (H.-J.B., R.B.).

## Author contributions

H.-J.B. proposed and supervised the work. X.L. prepared the samples, built up the setup, performed the tilted-plate experiments, analyzed, and explained the data under H.-J.B. and R.B.'s supervision. F.B. and M.Y. carried out the numerical diffuse-interface simulations of drop motion under H.M.'s supervision. All the authors discussed and interpreted the results. X.L. and H.-J.B. wrote the manuscript.

## Funding

## Competing interests

The authors declare no competing interests.
