## [Peer Review File · Nature Communications]

Kinetic Drop FrictionREVIEWER COMMENTS

Reviewer #1 (Remarks to the Author):

The paper shows that there is a linear relation between the force required to slide a drop and the product of the drop's speed, the viscosity and the drop's width. Yet extrapolating this linear relation to zero velocity intersects with a positive finite force. The paper explains the trend as a combination of two different different energy mechanisms: contact angle hysteresis/capillarity and viscous dissipation. The explanation is logic, and the paper is interesting. I recommend the paper for publication provided the following issues are addressed:

1. I very much appreciate table 1 in which one can find values of k_s that range between 0.43 to 1.46 – this is a very important finding that can be discussed, possibly in relation to the ratio h/A (height / area) that may come instead of w as discussed in Ref. 19. I recommend that the authors devote a paragraph to this possibility.
2. Viscous dissipation is known to be proportional to the sheared area. I, therefore, wonder if the proportionality to w in Eq. 2 can also be replaced by a proportionality to A/h .
3. There is a Nature Communications paper that discusses viscous dissipation of drops that I wonder of the authors are aware of: Timonen et al. "Free-decay and resonant methods for investigating the fundamental limit of superhydrophobicity", Nature Communications 2013. I recommend addressing it in the manuscript – I think they have relevant topics.
4. In the abstract, the text "we measured the velocity U , width w , length, advancing..." is a little inconsistent (either remove the U and the w symbolism, or add the symbolism for the length).
5. In fig. 1 there are blue vertical arrows. I recommend that the caption explains what they represent (I suppose this is the magnitude of the surface energy?).
6. In Fig 2d, the water on silicon data seem like a set of almost parallel lines as opposed to random noise. Is there a reason for that?
7. Is the "apparent capillary force" the same as the capillary force? Please clarify in the paper.
8. In figure 4g, the x axis does not start from zero. I suggest redrawing it in accordance to the other plots in that figure.

Reviewer #2 (Remarks to the Author):

The manuscripts examines the dynamics of liquid drops sliding down an incline. The authors have conducted a wide range of experiments to investigate the role of liquid viscosity and surface properties on friction coefficients. The investigation is conducted systematically and data are analyzed and interpreted carefully. While experimental results support the development of a comprehensive view on sliding droplets, the work does not offers significantly new physical insights and does not reveal novel interfacial fluid phenomena. Experimental techniques appear standard for this type of study and no fine data analysis is provided to clarify the wide spread of data points. Finally, the paper is generally well-written but style and figure quality remain average. In my view, the manuscript is more suited for a specialized journal, such as for instance JFM or PRFluids.

Reviewer #4 (Remarks to the Author):

I have attached a pdf copy of my report because it includes an equation.

Reviewer Report on Drop friction by Butt et al

This is an interesting and noteworthy report providing an extensive set of measurements on droplet friction for a range of liquids on smooth surfaces. It is topical in addressing questions about kinetic friction for droplets and the dependence of kinetic friction on speed of motion and viscosity of the liquid. Simulations are provided to complement the experimental work. The work uses a sound experimental methodology and is performed to a high standard within the field. Sufficient experimental information is included for the methods to be reproduced, and a useful supplementary information file is included. I believe this is an important and significant data set on an important topic and merits publication in *Nature Communications*. However, I have some reservations in how this work is currently presented and the strength of some conclusions.

The manuscript follows earlier work published in *Nature Physics* by some of the same authors about being able to think of droplet friction as having static and kinetic regimes in a similar manner to solids-sliding-on-solids (Gao, N.; Geyer, F.; Pilat, D. W.; Wooh, S.; Vollmer, D.; Butt, H.; Berger, R. "How Drops Start Sliding over Solid Surfaces". *Nature Physics* **2018**, *14* (2), 191–196). It also relates to work published by some of the authors on how droplet shape changes as a droplet moves from static to in-motion results in a maximum in the static friction (Laroche, A.; Naga, A.; Hinduja, C.; Sharifi, A. A.; Saal, A.; Kim, H.; Gao, N.; Wooh, S.; Butt, H.; Berger, R.; et al. "Tuning Static Drop Friction". *Droplet* **2023**, *2* (1), 1–8).

A key claim in this current manuscript is that the kinetic friction force can be empirically described by a constant force with an additional velocity dependent term scaling with droplet width and viscosity. The authors define a coefficient of proportionality for the velocity dependent term as a coefficient of friction. They also suggest that this coefficient is an independent material parameter characteristic specific to the liquid/surface combination and is neither dependent on the mean contact angle nor contact angle hysteresis. My substantive comments relate to these central parts of the paper. If the authors' accept these comments they will need to reconsider the abstract, particularly the final sentence, and conclusions.

Comments in Detail

1. The paper is focused on kinetic friction of droplets and not simply friction as might be presumed by some Readers from the title of the paper. I would suggest a refinement of the title of this paper.
2. The suggestion of a velocity dependent component with the dependencies suggested here is not new (but the data set establishing it is new and important).

Reference 12 was a reanalysis of the authors' *Nature Physics* data and included a definition of a coefficient of kinetic friction. To first order the coefficient was assumed constant (Coulomb's law/Amontons' third law), but ref. 12 also hypothesized a first order correction dependent on velocity (or Capillary number) using Cox-Voinov theory for dynamic contact angles. Taking eq. 20 from ref. 21 gives a term linear in viscosity, drop width and speed, i.e.

$$F_f = \left(\mu_s + \frac{6kCa}{\pi\theta_e^2} \log_e \left(\frac{L}{l_m} \right) \right) F_N = \mu_s F_N + \left[6k \left(\frac{\sin \theta_e}{\theta_e^2} \right) \log_e \left(\frac{L}{l_m} \right) \right] w\eta U \quad (R1)$$

This provides a prediction, which may or may not explain the material parameter specific to the liquid/surface combinations in the current manuscript. The authors' may wish to comment on an existing hypothesis for a velocity dependent component to the kinetic friction.

3. The authors' use of the term coefficient of friction for their constant of proportionality may introduce further fragmentation and confusion of notation and language into the literature. They are aware of this in their discussion of references 25, 26 and 12 below equation 2.

Based on comment 2 above they may wish to reconsider the terminology for what they define as a coefficient of friction and the use of the symbol μ . It also seems reasonable that in discussing ref. 12 in relation to their eq. 2 they should indicate that ref. 12 also provides a similar equation and has a prediction for how the term linear in velocity depends on the specific liquid/solid surface combination. I would also be cautious fig. 2e might be misinterpreted by Readers in comparison to work in ref. 12 on coefficients of friction.

4. The authors' state on page 6 that their coefficient of friction μ depends neither on the mean contact angle nor the contact angle hysteresis. Based on Fig. S26 I am not as convinced as them of this conclusion.

Fig. 26a has very scattered data points. However, there is a hint that it might reasonably be consistent with equation R1 with the exception of a cluster of three data points, which I think correspond to systems 21-23 (5, 10 and 50 cSt silicone oil droplets on Teflon). Silicone oil can be quite different to other liquids due to its strong spreading properties. I also note that glycerol can be very hygroscopic which has a large impact on viscosity estimates. Fig. 3 on page 7 of the manuscript also explains these were different to other systems in reaching their steady state velocity within the first mm of sliding. I have not checked absolute numbers for eq. R1, but I did plot the contact angle dependence on Fig. 26a.

I am also aware that eq. R1 assumes the systems are described by the hydrodynamic theory and that Molecular-Kinetic Theory (MKT) may give some different results. This may be relevant to the PDMS surface (see Barrio-Zhang, H.; Ruiz-Gutiérrez, É.; Armstrong, S.; McHale, G.; Wells, G. G.; Ledesma-Aguilar, R. Contact-Angle Hysteresis and Contact-Line Friction on Slippery Liquid-like Surfaces. *Langmuir* **2020**, *36* (49), 15094–15101).

The queries raised here are whether (i) the accuracy and scatter in fig. 26a allows a strong conclusion that the authors' μ parameter does not depend on the mean contact angle, and (ii) eq. R1 gives plausible orders of magnitude and contact angle trends.

5. The authors' include sentences as questions ending in "?".

This is a particular style, which is less usual in the literature.

6. Figure 1 has a typographical error in "resistance". There are also some typographical errors in the supplementary information (e.g. Fig. S1a).
7. I am less confident than the authors on the use of an effective mass in eq. 1. However, it does not introduce a large correction.
8. Water on grafted PDMS (system 4) has a large static contact angle hysteresis compared to what I might expect from the published work of Wang & McCarthy ("Covalently Attached Liquids: Instant Omniphobic Surfaces with Unprecedented Repellency". *Angew. Chemie Int. Ed.* **2016**, *55* (1), 244–248.). I wonder why a lower contact angle hysteresis grafted-PDMS surface was not used.
9. Fig. 3 has reference to the apparent capillary force, but how capillary force relates to fig. 3 is not well explained in the manuscript.

10. Below equation 5 the authors comment that their average k-factor is consistent with ElSherbini and Jacobi's calculation. It would be helpful to state explicitly that this is $24/\pi^3=0.774$.

11. Page 10 discusses why eq. 4 may be able to describe the kinetic force on sliding drops. The discussion at this point of the manuscript is quite qualitative. Equation R1 and the discussion in ref. 21 relating to the Cox-Voinov theory may provide an explanation.

I also wonder about literature on the friction from droplets moving on liquid-infused surfaces were both a linear relationship between friction and speed (a classical Stokes-type law), and a friction varying as a $2/3$ power law of speed have been discussed. I understand the current manuscript does not use liquid-infused surfaces and so the authors may regard this comment as outside the scope of the manuscript. However, some of the discussion about friction laws may be of interest. Relevant literature here includes a paper by one of the current co-authors,

Keiser, A.; Baumli, P.; Vollmer, D.; Quéré, D. "Universality of Friction Laws on Liquid-Infused Materials". *Phys. Rev. Fluids* **2020**, 5 (1), 014005.

and

Smith, J. D.; Dhiman, R.; Anand, S.; Reza-Garduno, E.; Cohen, R. E.; McKinley, G. H.; Varanasi, K. K. "Droplet Mobility on Lubricant-Impregnated Surfaces". *Soft Matter* **2013**, 9 (6), 1772–1780.

We thank all the reviewers for their valuable comments and feedback! We have modified our manuscript accordingly. The major changes are:

1. Comparison with the existing hydrodynamic hypothesis.
2. Discussion on other scaling possibilities for the empirical law.
3. Comments on the existing methods, explanations, laws, and coefficient of drop friction.

We believe that addressing these points has substantially improved our paper. In the following, we respond to the referees' comments in detail. Our answers are marked in blue and changed text passages appear in purple.

To reviewer #1:

The paper shows that there is a linear relation between the force required to slide a drop and the product of the drop's speed, the viscosity, and the drop's width. Yet extrapolating this linear relation to zero velocity intersects with a positive finite force. The paper explains the trend as a combination of two different energy mechanisms: contact angle hysteresis/capillarity and viscous dissipation. The explanation is logic, and the paper is interesting. I recommend the paper for publication provided the following issues are addressed:

1. I very much appreciate Table 1 in which one can find values of k_s that range between 0.43 to 1.46 – this is a very important finding that can be discussed, possibly in relation to the ratio h/A (height/area) that may come instead of w as discussed in Ref. 19. I recommend that the authors devote a paragraph to this possibility.

Thanks a lot for the stimulating remark! In order to maximize data utilization, we will deposit all the related datasets supporting the figures and tables in this manuscript.

Our present setup cannot record the sliding drop from the bottom view to measure the contact area directly. For this reason, we estimated the ratio h/A from the mean contact angle θ_m (the average of static advancing and receding contact angle) and the drop volume V . By assuming the static drop is a spherical cap with a contact radius r , we estimated the drop height by $r(1 - \cos\theta_m)/\sin\theta_m$ and the contact area A by πr^2 . Knowing the drop volume V , the radius r can be

calculated by $r = \sqrt[3]{\frac{3V \sin^3 \theta_m}{\pi(1 - \cos \theta_m)^2(2 + \cos \theta_m)}}$. Then we plotted k_s -versus- h/A in revised Fig. S26.

However, we didn't find a strong correlation between k_s and h/A .

In the revised version, we have added a discussion on page 10. It reads: "As an alternative, Tadmor proposed to use the length prefactor $L_p = A/h$ rather than the width of the drop to calculate the friction force of a sliding drop. Here, A is the contact area of the drop, and h is its height²⁴. Estimating the correlation between L_p and the k_s factor led, however, not to a significant correlation (Fig. S26)."

Figure S26. The correlation between static k -factor (k_s) and h/A (h is the height of the drop while A is the liquid-solid contact area). (a) Schematics of a spherical cap. By assuming the static drop is a spherical cap with contact radius r , we estimated the drop height by $h = r(1 - \cos \theta_m)/\sin \theta_m$ ($\theta_m = \frac{\theta_{as} + \theta_{rs}}{2}$ is the mean contact angle) and the contact area by $A = \pi r^2$.

Knowing the drop volume V , the contact radius was calculated with $r = \sqrt[3]{\frac{3V \sin^3 \theta_m}{\pi(1 - \cos \theta_m)^2(2 + \cos \theta_m)}}$.

Finally, we calculated h/A based on the mean contact angle and drop volume. (b) The plot of k_s versus h/A .

2. Viscous dissipation is known to be proportional to the sheared area. I, therefore, wonder if the proportionality to w in Eq. 2 can also be replaced by a proportionality to A/h .

Thanks for the inspiring remark! In a planar Couette flow, the flow velocity varies linearly from bottom to top, the viscous force is indeed found to be proportional to the shear area. But for a

sliding drop, the simulation results in Fig. 5 show that viscous dissipation density is significantly higher in the wedge than in the bulk. Over 50% of viscous dissipation happens near the contact line. Thus, it is not immediately clear whether the viscous dissipation is proportional to the contact area. Because our present setup cannot directly measure the contact area of sliding drops and the contact area changes significantly with velocity, which is different from the case of static drops. To find out whether a scaling with A/h leads to a better scaling, we would need precise measurements of the contact area. Because for a spherical-cap drop shape, A/h is proportional to w ($A/h = \pi r \frac{\sin\theta_m}{1-\cos\theta_m}$, and $r=w/2$). Thus, we are looking for a second-order correction. But we agree that it would be interesting to check the possibility of replacing w with A/h in the future or see how relevant the length of the contact line is for energy dissipation.

3. There is a Nature Communications paper that discusses viscous dissipation of drops that I wonder of the authors are aware of: Timonen et al. "Free-decay and resonant methods for investigating the fundamental limit of superhydrophobicity", Nature Communications 2013. I recommend addressing it in the manuscript – I think they have relevant topics.

Thanks for pointing out the relevant literature! In Jaakko Timonen's paper, the viscous force increases with the normal force and is nearly linearly proportional to the contact area for oscillated magnetic water drops on superhydrophobic surfaces. The situation is, however, quite different from our experiment. First, their superhydrophobic surfaces have higher apparent contact angles than flat surfaces. The higher the apparent contact angle, the lower the viscous dissipation near the contact line. Second, in contrast to our flat surfaces, superhydrophobic surfaces show apparent slip. This different boundary condition changes the flow pattern and viscous dissipation. Third, the dispersed iron oxide particles in Timonen's experiments influence the flow pattern inside the drop. For these reasons, a direct comparison may be difficult. We appreciate, however, that the method is quite important to measure energy dissipation.

In the revised main text, we have added a discussion on page 2 and have included the reference. It reads: "There are many methods to study drop dynamics, such as using de-/inflated drops⁹, magnetically controlled oscillated drops¹⁰, direct force measurements with force sensors^{11, 12}, and sliding drops on a tilted surface¹³." [New ref. 10: Timonen, J. V., Latikka, M., Ikkala, O., Ras,

R. H. Free-decay and resonant methods for investigating the fundamental limit of superhydrophobicity. Nature Communications, 4(1), 2398 (2013).]

4. In the abstract, the text "we measured the velocity U , width w , length, advancing..." is a little inconsistent (either remove the U and the w symbolism or add the symbolism for the length.

Thanks for the reminder! Indeed, there should be a symbol L in this sentence. We have added it to the revised abstract.

5. In Fig. 1 there are blue vertical arrows. I recommend that the caption explains what they represent (I suppose this is the magnitude of the surface energy?).

Thanks for the helpful suggestion! The blue vertical arrows indicate the possible energy dissipation due to surface adaptation, corresponding to the details in the blue box underneath. To make the figure more understandable, in the revised version, we have color-matched the arrows and the boxes below and have revised the caption of Fig. 1 as "Schematic of all the possible effects leading to drop friction. The arrows indicate the corresponding effects in the boxes below (yellow: viscous dissipation; orange: contact-line friction induced by molecular kinetics, defects, or deformation; blue: surface adaptation; green: slide electrification, black: air resistance)."

6. In Fig 2d, the water on silicon data seems like a set of almost parallel lines as opposed to random noise. Is there a reason for that?

This is an important point that we have to clarify. The parallel data points of F or F/w occur when the drop slides with steady-state velocity (no acceleration) at each titled angle like the case shown in Fig. 2b. Based on equation (1), $F_f = mgs\sin\alpha - m^* \frac{dU}{dt}$, the force is almost constant (parallel data) when acceleration (dU/dt) is zero. Thus, from Fig. 3, as a "zoom-in" figure of Fig. 2d, we can observe this for most liquid drops with low surface tension and high viscosity, which typically slide with low velocity and reach their steady-state velocities within a few mm sliding. This has been explained in the caption of Fig. 3 with "In some cases like 60-95% glycerol-water mixture or silicone oil, the drop reached its steady-state velocity within the first mm of sliding so that $dU/dt \approx 0$. As a result, the graph $F_f/w - vs - U\eta$ looks non-continuous. In contrast, for

low-viscosity liquids such as water, drops accelerated within the whole recorded slide length, leading to continuous $F_f/w - vs - U\eta$ graphs.” In the revised version, we have added a connection between Fig. 2d and Fig. 3 in the caption of Fig. 2d with “The details and zoom-in of most curves refer to Fig. 3.”

7. Is the "apparent capillary force" the same as the capillary force? Please clarify in the paper.

Sorry about the confusion! We often use the term “apparent” capillary force to point out that apparent contact angles are inserted in Eq. (3) and Eq. (4).” To avoid any confusion, we have removed “apparent” and only used the concept of “capillary force” in the revised version.

8. In Figure 4g, the x-axis does not start from zero. I suggest redrawing it in accordance with the other plots in that figure.

Thanks for the reminder, we have replotted Fig. 4g with an x-axis starting from 0 in the revised version.

To reviewer #2:

The manuscript examines the dynamics of liquid drops sliding down an incline. The authors have conducted a wide range of experiments to investigate the role of liquid viscosity and surface properties on friction coefficients. The investigation is conducted systematically and data are analyzed and interpreted carefully. While experimental results support the development of a comprehensive view on sliding droplets, the work does not offer significantly new physical insights and does not reveal novel interfacial fluid phenomena. Experimental techniques appear standard for this type of study and no fine data analysis is provided to clarify the wide spread of data points. Finally, the paper is generally well-written but style and figure quality remain average. In my view, the manuscript is more suited for a specialized journal, such as for instance JFM or PRFluids.

Thanks for the critical comments! The main contribution of the manuscript is offering an empirical law of drop friction and the liquid-surface properties dependent friction coefficient for drop-sliding prediction. In addition, as new physical insights, we clarify which dissipation channels contribute to drop friction and how much they contribute. This we do by experiments and with simulations.

Stimulated by the comment, we compared our experimental results with predictions of hydrodynamic theory. Usually, it is assumed that the main energy dissipation of fast drops is viscous dissipation near the contact line. We measured the kinetic advancing and receding contact angles for 23 liquid/solid combinations. For water drops, fitting the receding contact angles, $\theta_r(U)$, with hydrodynamic theory results in unrealistic fitting parameters. Therefore, we concluded that additional energy dissipation processes contribute substantially to drop friction for water. We have added this new physical insight into the revised main text with a new section named "Contribution of viscous force to drop friction".

We have published the experimental technique to determine drop friction by applying the equation of motion (Li *et al.* Nature Physics 2022, 18, 713). It was applied to determine electrostatic effects. Here, it is used for the first time to analyze drop friction in general. It opens the possibility to analyze drop friction at a large range of velocities. Before, methods were restricted to low velocities and/or low-friction surfaces (superhydrophobic). Though measuring

sliding drops on tilted surfaces with a high-speed camera is not new, it is still a challenge to measure contact-line velocity, kinetic advancing contact angle, kinetic receding contact angles, contact length, and contact width simultaneously. We succeed to do so. We offer datasets with a wide range of velocity from 10^{-5} – 0.7 m/s to the community. Future research in the related fields will also benefit from this rich database. The widespread data points are the raw data that we measured directly by experiments, which are clear enough to support our arguments in this manuscript.

In contrast to the earlier work (Li *et al.* Nature Physics 2022, 18, 713), we deliberately reduced electrostatic effects by taking conducting or high-permittivity substrates.

To reviewer #4:

This is an interesting and noteworthy report providing an extensive set of measurements on droplet friction for a range of liquids on smooth surfaces. It is topical in addressing questions about kinetic friction for droplets and the dependence of kinetic friction on speed of motion and viscosity of the liquid. Simulations are provided to complement the experimental work. The work uses a sound experimental methodology and is performed to a high standard within the field. Sufficient experimental information is included for the methods to be reproduced, and a useful supplementary information file is included. I believe this is an important and significant data set on an important topic and merits publication in Nature Communications. However, I have some reservations in how this work is currently presented and the strength of some conclusions.

The manuscript follows earlier work published in Nature Physics by some of the same authors about being able to think of droplet friction as having static and kinetic regimes in a similar manner to solids-sliding-on-solids (Gao, N.; Geyer, F.; Pilat, D. W.; Wooh, S.; Vollmer, D.; Butt, H.; Berger, R. “How Drops Start Sliding over Solid Surfaces”. Nature Physics 2018, 14 (2), 191–196). It also relates to work published by some of the authors on how droplet shape changes as a droplet moves from static to in-motion results in a maximum in the static friction (Laroche, A.; Naga, A.; Hinduja, C.; Sharifi, A. A.; Saal, A.; Kim, H.; Gao, N.; Wooh, S.; Butt, H.; Berger, R.; *et al.* “Tuning Static Drop Friction”. Droplet 2023, 2 (1), 1–8).

Thanks for the stimulating comments! In fact, the results reported here were motivated by these earlier experiments. The earlier measurements were carried out by the drop adhesion force instrument and limited to slow velocities (≈ 1 cm/s). To see the influence of viscous dissipation for liquids like water, one needs to go to high velocity. Using the tilted plate and analyzing the equation of motion made such measurements possible. To make the connection between the present work to the previous work clear, we have added a comment in the second paragraph on page 2. It reads: “There are many methods to study drop dynamics, such as using de-/inflated drops⁹, magnetically controlled oscillated drops¹⁰, direct force measurements with force sensors^{11, 12}, and sliding drops on a tilted surface¹³.” [New ref. 11 & 12: Gao, N.; Geyer, F.; Pilat, D. W.; Wooh, S.; Vollmer, D.; Butt, H.; Berger, R. How Drops Start Sliding over Solid Surfaces. *Nature Physics* 2018, 14 (2), 191–196. / Laroche, A.; Naga, A.; Hinduja, C.; Sharifi, A. A.; Saal, A.; Kim, H.; Gao, N.; Wooh, S.; Butt, H.; Berger, R.; et al. Tuning Static Drop Friction. *Droplet* 2023, 2 (1), 1–8]

A key claim in this current manuscript is that the kinetic friction force can be empirically described by a constant force with an additional velocity-dependent term scaling with droplet width and viscosity. The authors define a coefficient of proportionality for the velocity-dependent term as a coefficient of friction. They also suggest that this coefficient is an independent material parameter characteristic specific to the liquid/surface combination and is neither dependent on the mean contact angle nor contact angle hysteresis. My substantive comments relate to these central parts of the paper. If the authors accept these comments, they will need to reconsider the abstract, particularly the final sentence, and conclusions.

1. The paper is focused on the kinetic friction of droplets and not simply friction as might be presumed by some Readers from the title of the paper. I would suggest a refinement of the title of this paper.

This is a valid point. The new experimental results are indeed on kinetic friction. We also draw some conclusions about static friction (onset of sliding). By comparing the capillary force calculated by eq. (4) and the friction force at zero velocity calculated by eq. (2), we determine the static k -factor, whose value is still an open question in this field. For this reason, we thought

the title needed to be justified. To emphasize the new database, which is on kinetic friction, we have replaced our title with: “Kinetic Drop Friction”.

2. The suggestion of a velocity-dependent component with the dependencies suggested here is not new (but the data set establishing it is new and important).

Reference 12 was a reanalysis of the authors’ Nature Physics data and included a definition of a coefficient of kinetic friction. To first order, the coefficient was assumed constant (Coulomb’s law/Amontons’ third law), but ref. 12 also hypothesized a first-order correction dependent on velocity (or Capillary number) using Cox-Voinov theory for dynamic contact angles. Taking eq. 20 from ref. 12 gives a term linear in viscosity, drop width, and speed, i.e.

$$F_f = \left(\mu_s + \frac{6kCa}{\pi\theta_e^2} \log_e \left(\frac{L}{l_m} \right) \right) F_N = \mu_s F_N + [6k \left(\frac{\sin\theta_e}{\theta_e^2} \right) \log_e \left(\frac{L}{l_m} \right)] w\eta U \quad (\text{R1})$$

This provides a prediction, which may or may not explain the material parameter specific to the Liquid/surface combinations in the current manuscript. The authors may wish to comment on an existing hypothesis for a velocity-dependent component to kinetic friction.

Thanks for the inspiring comments. Yes, we should have included such a discussion! We did three things. First, we fitted the measured kinetic contact angles with the Cox-Voinov theory: $\theta_{a/r} = \left(\theta_{a0/r0}^3 \pm 9Ca \ln \frac{l}{l_m} \right)^{1/3}$ and added the fit in revised Fig. 4e and Fig. S4-25c. The fitting parameters including the ratio of the macroscopic-to-microscopic length scale, l/l_m , and the microscopic contact angle, $\theta_{a0/r0}$ have been listed in Table S1 in the revised supporting information. Fitting of the receding contact angle for water turned out to lead to unrealistic values for l/l_m . Even then, in some cases, the fit is not good. We conclude that for water, viscous dissipation alone cannot explain the changes in kinetic contact angles. Other effects contribute substantially.

Second, we calculated the friction force versus capillary number Ca assuming that the change in contact angles is given by viscous dissipation near the contact line, $\theta_{a/r} = \left(\theta_{a0/r0}^3 \pm 9Ca \ln \frac{l}{l_m} \right)^{1/3}$. Graphs obtained with these contact angles inserted into eq. (4), $F_c =$

$k\omega\gamma(\cos \theta_r - \cos \theta_a)$, are plotted in revised Fig. S27a. The hydrodynamic contribution can be well approximated as being linear up to Ca of the order of 0.01-0.02.

Third, as the reviewer's comment 4 below, we derived an equation equivalent to eq. (20) in Ref. 12 [McHale *et al.*, Langmuir 2022, 38, 4425]. Then the friction coefficient should be given by $\beta' = 3k \ln \frac{l}{l_m} \cdot \left[\frac{\sin \theta_{a0}}{\theta_{a0}^2} + \frac{\sin \theta_{r0}}{\theta_{r0}^2} \right]$, assuming that hydrodynamic friction near the contact lines dominates. When the fitting parameter $(l/l_m)_{a/r}$ (listed in Table S1) and $\theta_{a0/r0}$ are inserted, the resulting friction coefficient is comparable to our empirical value. However, our empirical friction coefficient is two times higher than the μ_k predicted with eq. (R1) above when reasonable $\left(\frac{l}{l_m}\right) = 10^4, 10^6, \text{ or } 10^8$ and mean contact angle are inserted (revised Fig. S27b). This finding confirms that viscous dissipation near the contact line alone is not sufficient to account for drop friction. In addition, we did not find a strong correlation between our empirical friction coefficient and the mean contact angle (Fig. S27b).

In the revised main text, we have added the conclusion to the abstract with "For water, viscous dissipation alone cannot explain the changes in kinetic contact angles. Other effects contribute substantially." and to the conclusion section with: "For water, viscous dissipation alone cannot explain the changes in kinetic contact angles. Other dissipation channels contribute substantially."

We have also added a new section in the main text. It reads:

"Contribution of viscous force to drop friction

The above-mentioned proportionality of the friction force to velocity and viscosity indicates that hydrodynamic dissipation plays a substantial role. For this reason, it is not a priori clear why equation (4) is able to describe the kinetic friction force of sliding drops. A possible explanation is that viscous dissipating processes influence the contact angles, which enter equation (4). With our setup, we measured the contact angles on a length scale of 10-100 μm . Energy dissipation occurring closer than 100 μm to the contact line is taken into account. Thus, wedge viscous dissipation has been included and should show up as a change in contact angle and via equation (4) in friction (Fig. 4j).

Viscous dissipation brought by shear flow near the contact line has been modeled by Cox, Voinov, and others using continuum hydrodynamics⁴⁵⁻⁴⁷. To check whether the change in contact

angle is *only* due to viscous dissipation, we fitted the velocity-dependent kinetic contact angles with the Cox-Voinov theory^{44, 45}:

$$\theta_{a/r}(U) = \left(\theta_{a0/r0}^3 \pm 9Ca \ln \frac{l}{l_m} \right)^{1/3} \quad (6)$$

Here, $\theta_{a0/r0}$ are the microscopic advancing and receding contact angles. l is a macroscopic cut-off length related to the drop size. l_m is a microscopic cut-off length, usually of molecular scale, below which slip is allowed. With $l \approx 1$ mm and $l_m \approx 1$ nm, a ratio of the order of $l/l_m \approx 10^6$ is expected. The fitting of the receding contact angle for water leads to 10 orders of magnitude higher values for l/l_m (Fig. 4g, Fig. S4-S25c, Table S1). We conclude that for water, viscous dissipation cannot explain the entire velocity-dependent friction.

To check for the other liquids whether the linear kinetic friction force is only due to viscous force, we calculated the friction force versus capillary number Ca assuming that the change in contact angles is given by equation (6). It turns out that the hydrodynamic contribution to drop friction can be well approximated as being linear up to Ca of the order of 0.01-0.02 (Fig. S27a). McHale *et al.* already considered the linear part of F_c -vs- Ca graphs by inserting expression (6) into equation (4) and taking the first element of a Taylor series¹⁷. In analogy with their approach, by setting friction force in equation (2) equal to the capillary force in equation (4), we obtain:

$$\beta' = 3k \ln \frac{l}{l_m} \cdot \left[\frac{\sin \theta_{a0}}{\theta_{a0}^2} + \frac{\sin \theta_{r0}}{\theta_{r0}^2} \right] \quad (7)$$

The superscript in β' is to distinguish from the empirical β listed in Table 1. When estimating the friction coefficient with typical parameters for water-glycerol mixtures ($\theta_{a0} = \theta_{r0} = 100^\circ$, $l/l_m = 10^6$, $k=1$), we get $\beta' \approx 22$. For silicon oils ($\theta_{a0} = \theta_{r0} = 50^\circ$, $l/l_m = 10^6$, $k=1$), we get $\beta' \approx 70$. The order of magnitude agrees with experimental values (Table 1). We did not find a significant correlation between β and the contact angle due to the scattering data points (Fig. S27b)."

In the revised supporting information, we have added a table S1 and Fig.S17. They read:

Table S1. Fitting parameters (θ_{a0} , θ_{r0} , and $\frac{l}{l_m}$) when fitting the velocity-dependent kinetic contact angles by $\theta_{a/r} = \left(\theta_{a0/r0}^3 \pm 9 \frac{U\eta}{\gamma} \ln \frac{l}{l_m} \right)^{1/3}$ (Cox-Voinov theory) in Fig. 4d and Fig. S4-25c.

	Liquid-Surface	γ	η	$\theta_{a0}^{(1)}$	$\theta_{r0}^{(1)}$	$(l/l_m)_a^{(2)}$	$(l/l_m)_r^{(2)}$

		mN/m	$mPa \cdot s$	°	°	(Advancing)	(Receding)
1	Water-Si wafer	72	0.92	44	26	1.1×10^{24}	1.5×10^7
2	Water-ITO glass	72	0.92	118	91	2.4×10^3	5.2×10^{23}
3	Water-PFOTS	72	0.92	106	90	3.3×10^{14}	1.9×10^{18}
4	Water-PDMS	72	0.92	110	91	9.0×10^{12}	1.2×10^{26}
5	Water-PS	72	0.92	95	82	2.5×10^8	5.2×10^{17}
6	Water-Thiols	72	0.92	132	107	4.3×10^{10}	2.9×10^{26}
7	Water-Teflon	72	0.92	125	119	2.8×10^6	1.5×10^{25}
8	30% Glycerol-Teflon	69	2.5	119	112	8.0×10^4	2.1×10^{15}
9	40% Glycerol-Teflon	69	3.8	117	108	7.8×10^4	3.2×10^{10}
10	50% Glycerol-Teflon	68	6.9	117	108	2.2×10^4	1.2×10^8
11	60% Glycerol-Teflon	67	13.6	122	108	1.2×10^3	6.7×10^5
12	70% Glycerol-Teflon	66	27.1	120	104	5.5×10^3	6.0×10^5
13	80% Glycerol-Teflon	66	75.9	118	102	2.2×10^3	1.7×10^4
14	85% Glycerol-Teflon	65	93	116	103	8.9×10^4	2.1×10^6
15	90% Glycerol-Teflon	65	192	116	102	4.6×10^4	2.2×10^6
16	95% Glycerol-Teflon	65	265	119	105	9.3×10^5	1.6×10^{11}
17	99% Glycerol-Teflon	64	943	118	102	4.6×10^1	5.6×10^3
18	Ethylene glycol-Teflon	48	16	100	92	7.2×10^5	1.5×10^{10}
19	Formamide-Teflon	58	4.6	115	103	7.5×10^3	6.1×10^7
20	Ionic liquid-Teflon	51	22	103	91	1.1×10^5	3.1×10^7
21	5 cSt silicone oil-Teflon	21	5	46	41	2.8×10^9	1.3×10^5
22	10 cSt silicone oil-Teflon	21	10	54	46	3.4×10^2	1.5×10^6
23	50 cSt silicone oil-Teflon	21	50	57	50	9.4×10^2	2.0×10^4

Note: (1) θ_{a0} and θ_{r0} are microscopically advancing and receding contact angles in the Cox-Voinov theory, different from static advancing and receding contact angles ($\theta_{as/rs}$) in Table 1. (2) l/l_m is the ratio of macroscopic length scale (l) to microscopic length (l_m) scale. We fitted the velocity-dependent advancing and receding contact angles separately, therefore, $\frac{l}{l_m}$ for the advancing and receding sides is different.

Figure S27. Hydrodynamic hypothesis. (a) Capillary forces scaled by the width of the drop w , the surface tension of the liquid, and the geometry factor k ($\frac{F_c}{k\gamma w} = \cos\theta_r - \cos\theta_a$) versus capillary number (Ca). By assuming $\theta_{a/r} = \left(\theta_{a0/r0}^3 \pm 9\frac{U\eta}{\gamma} \ln\frac{l}{l_m}\right)^{1/3}$, we calculated $\cos\theta_r - \cos\theta_a$ with different ratios of the macroscopic-to-microscopic length scale, l/l_m . (b) The correlation between friction coefficient and contact angle. The solid curves present the theoretical correlation between the mean contact angle (θ_m) and friction coefficient base on $\beta' = 6k \ln\frac{l}{l_m} \cdot \left(\frac{\sin\theta_m}{\theta_m^2}\right)$ when inserting $l/l_m = 10^4$ (green), 10^6 (blue), and 10^8 (orange).

3. The authors' use of the term coefficient of friction for their constant of proportionality may introduce further fragmentation and confusion of notation and language in the literature. They are aware of this in their discussion of references 25, 26, and 12 below Equation 2.

Based on comment 2 above they may wish to reconsider the terminology for what they define as a coefficient of friction and the use of the symbol μ . It also seems reasonable that in discussing ref. 12 in relation to their eq. 2 they should indicate that ref. 12 also provides a similar equation and has a prediction for how the term linear in velocity depends on the specific liquid/solid

surface combination. I would also be cautious fig. 2e might be misinterpreted by Readers in comparison to the work in ref. 12 on coefficients of friction.

This is a valid point. We agree with the reviewer's comment about the friction coefficient. In the revised version, we have replaced the symbol for friction coefficient with β to avoid confusion. To clarify the difference in the definition of friction coefficients, we have added the statement "The definition of friction coefficient here is different from previous ones in literature^{17, 29, 30}." in the caption of Fig. 2e and Table 1. We have also listed all reported symbols of friction coefficient with their definition on page 6, which reads: "The terminology of friction coefficient here is different from Bocquet and Barrat's²⁹, de Ruijter's³⁰, or McHale's definition¹⁷. Bocquet and Barrat defined a phenomenological friction coefficient with the symbol λ as the ratio of friction force to velocity by the hydrodynamic approach for the liquid/solid boundary with a unit of Ns/m²⁹. de Ruijter defined a friction coefficient per unit length of contact line with the symbol ζ_0 based on the molecular kinetic theory for drop spreading in the unit of Pa · s³⁰. Similar to the solid/solid system, McHale defined the ratio of drop friction to its normal adhesion as friction coefficient, μ ¹⁷."

In addition, to compare with the predicted kinetic friction coefficient in reference 12, we have added a discussion on page 13. The predicted kinetic friction coefficient in reference 12 is much lower than our empirical friction coefficient when reasonable $\frac{l}{l_m}$ are inserted. The details are in the response to comment 2.

4. The authors' state on page 6 that their coefficient of friction μ depends neither on the mean contact angle nor the contact angle hysteresis. Based on Fig. S26, I am not as convinced as them of this conclusion.

Fig. 26a has very scattered data points. However, there is a hint that it might reasonably be consistent with equation R1 with the exception of a cluster of three data points, which I think correspond to systems 21-23 (5, 10, and 50 cSt silicone oil droplets on Teflon). Silicone oil can be quite different to other liquids due to its strong spreading properties. I also note that glycerol can be very hygroscopic which has a large impact on viscosity estimates. Fig. 3 on page 7 of the manuscript also explains these were different to other systems in reaching their steady-state

velocity within the first mm of sliding. I have not checked absolute numbers for eq. R1, but I did plot the contact angle dependence on Fig. 26a.

I am also aware that eq. R1 assumes the systems are described by the hydrodynamic theory and that Molecular-Kinetic Theory (MKT) may give some different results. This may be relevant to the PDMS surface (see Barrio-Zhang, H.; Ruiz-Gutiérrez, É.; Armstrong, S.; McHale, G.; Wells, G. G.; Ledesma-Aguilar, R. Contact-Angle Hysteresis and Contact-Line Friction on Slippery Liquid-like Surfaces. *Langmuir* 2020, 36 (49), 15094–15101).

The queries raised here are whether (i) the accuracy and scatter in Fig. 26a allows a strong conclusion that the authors' μ parameter does not depend on the mean contact angle, and (ii) eq. R1 gives plausible orders of magnitude and contact angle trends.

Thanks for pointing out that the statement “friction coefficient depends neither on mean contact angle nor contact angle hysteresis” based on scattered data points in Fig. S26 is too strong. We totally agree and have re-evaluated the correlation between our empirical friction coefficients with the mean contact angle. Details in the response to comment 2. The corresponding conclusion has also been revised as: “In addition, we did not find a correlation between β and the contact angle due to the scattering data points (Fig. S27b).” on page 13.

In addition, thanks for pointing out the relevant reference which reports contact-line friction on PDMS surfaces. We have cited it as a new reference 48 when discussing other dissipation channels than viscous dissipation on page 13. It reads: “These above results indicate that viscous dissipation near the contact line is not the only dissipation channel changing the kinetic contact angles, at least not for water. Only viscous dissipation is insufficient to account for drop friction. Other effects such as contact-line friction^{20, 48} or adaptation²³ contribute substantially.” [New ref. 48: Barrio-Zhang, H.; Ruiz-Gutiérrez, É.; Armstrong, S.; McHale, G.; Wells, G. G.; Ledesma-Aguilar, R. Contact-Angle Hysteresis and Contact-Line Friction on Slippery Liquid-like Surfaces. *Langmuir* 2020, 36 (49), 15094–15101]

5. The authors include sentences as questions ending in “?”. This is a particular style, which is less usual in the literature.

Thanks for the suggestion. We have revised the sentences ending in explicit question marks on page 3. Now they read: “The questions addressed were: How the drop friction depends on the velocity; which material parameters influence drop friction; How to describe friction forces quantitatively; which dissipation processes contribute how strongly to the friction of sliding drops. The aim is to predict drop sliding velocity quantitatively.”

6. Figure 1 has a typographical error in “resistance”. There are also some typographical errors in the supplementary information (e. g. Fig. S1a).

Thanks for the helpful reminder, we have checked and revised all the typos in the figure, the text, and the supporting information.

7. I am less confident than the authors on the use of an effective mass in eq. 1. However, it does not introduce a large correction.

We fully agree with the reviewer that the influence of effective mass should be considered carefully. We use effective mass for the consideration of the rolling component in the sliding drop when applying the no-slip boundary condition. The effective mass is determined by direct numerical simulation and defined by $m^* = 2E_{kin}/U^2$, in which E_{kin} is the kinetic energy of the drop and U is the drop velocity. For a water drop with a radius of 2.5 mm and an equilibrium contact angle of 90° , our previous simulated results indicated m^*/m increases from 1 to ~ 1.15 when velocity increases from 0 to 0.5 m/s at the tilted angle of 30° , 40° , and 60° [Fig. S7 in Ref.21: Spontaneous charging affects the motion of sliding drops. Nature Physics 18, 1-7 (2022)]. Therefore, we discussed the deviation between $m^*/m=1$ and $m^*/m= 1.1$ in Fig. S2. The deviation is around 1% for water drops on the PS-gold surface ($\theta \approx 90^\circ$) and 2.5% for 30% glycerol aqueous drops on the Teflon-gold surface ($\theta \approx 120^\circ$), which is lower than the variation of velocity observed from sample to sample. For other liquids with high viscosity and low surface tension, drops slide nearly with steady-state velocity (or zero acceleration), the selection of m^* makes no difference in force based on equation (1).

To make the discussion more comprehensive and further support our argument. In the revised version, we have added the deviation between $m^*/m=1$ and $m^*/m= 1.2$ as well (Green data point

in the revised Fig. S2). The average standard deviation is still low, 2.3% for water drops on the PS-gold surface and 5% for 30% glycerol aqueous drops on the Teflon-gold surface.

Figure S2. Deviation of friction force when m^*/m changes from 1.0 to 1.2. The friction forces deviate within 2.3% for the water drop on the PS-gold surface and within 5% for the 30% glycerol-water mixture on the Teflon-gold surface when m^*/m varies from 1.0 to 1.2. The deviation is calculated by $\frac{2(F_f - F'_f)}{(F_f + F'_f)} \times 100\%$ (F_f and F'_f are the force when $\frac{m^*}{m} = 1$ and $\frac{m^*}{m} \neq 1$ correspondingly).

8. Water on grafted PDMS (system 4) has a large static contact angle hysteresis compared to what I might expect from the published work of Wang & McCarthy (“Covalently Attached Liquids: Instant Omnipophobic Surfaces with Unprecedented Repellency”. *Angew. Chemie Int. Ed.* 2016, 55 (1), 244–248.). I wonder why a lower contact angle hysteresis grafted-PDFMS surface was not used.

We agree with the reviewer that our reported contact angle hysteresis (CAH) of the grafted PDMS is higher than the reported CAH of the grafted PDFMS. But it is similar to the reported CAH in ref 52. The higher CAH of our grafted PDMS surface could be due to: (1) different preparation methods. We prepared grafted PDMS brushes surfaces by the “graft to” method using poly(dimethylsiloxane) with a molecular weight of 6 kg/mol; Wang & McCarthy prepared grafted PDFMS surfaces by the “graft from” method using an isopropanol solution of $\text{Me}_2\text{Si}(\text{OMe})_2$. Different preparation methods might lead to different surface morphology and therefore

different contact angle hysteresis. (2) Different cleaning methods. To ensure the samples had reached a stable state before drop-sliding measurements, we cleaned the samples by ultrasonic in the bath of toluene, ethanol, and water for 10 min each, which is different from the rinsing method used by Wang & McCarthy. Because the preparation and related chemicals have been used in our lab and our previous publications. For easy comparison, we used the PDMS surface prepared with the “graft to” method again here. We appreciate Wang & McCarthy’s effort to reduce contact angle hysteresis by a novelty surface, but persuading an extremely low CAH is out of the scope of this manuscript.

To clarify the difference, we have added a statement in the section of Materials and Methods on page 17 and cited the related papers. The statement reads: “The ‘graft to’ method used here to prepare PDMS brushes is different from the ‘graft from’ method to prepare PDMS brushes⁵³ or PDFMS brushes⁵⁴, which might lead to different contact angle hysteresis.” [New ref. 54: Wang & McCarthy. Covalently Attached Liquids: Instant Omniphobic Surfaces with Unprecedented Repellency. *Angew. Chemie Int. Ed.* 2016, 55 (1), 244–248.]

9. Fig. 3 has reference to the apparent capillary force, but how capillary force relates to fig. 3 is not well explained in the manuscript.

Thanks for the reminder. In the revised version, we have referred to Fig.3 on page 10 “The capillary forces as given by equation (4) are a good estimate for measured kinetic friction forces for all tested liquids and surfaces (Fig 3, Fig. 4h, and Fig S4-S25).” In fact, Fig. 3 is also a zoom-in version of Fig. 2d. To clarify this, we also added a statement in the caption of Fig. 2d in the revised version. The statement reads: “The details and zoom-in of most curves refer to Fig. 3.”

10. Below equation 5 the authors comment that their average k-factor is consistent with ElSherbini and Jacobi’s calculation. It would be helpful to state explicitly that this is $24/\pi^3=0.774$. Thanks for the helpful reminder. We have revised this sentence on page 9. Now it reads: “The average *k*-factor of all the liquid/surface systems was 0.88 ± 0.2 for the onset of sliding, similar to ElSherbini and Jacobi’s calculation ($k=24/\pi^3=0.774$)³³ and Extrand’s experimental results ($k=4/\pi$ when using drop radius instead of drop width)^{38, 39}.”

11. Page 10 discusses why eq. 4 may be able to describe the kinetic force on sliding drops. The discussion at this point of the manuscript is quite qualitative. Equation R1 and the discussion in ref. 12 relating to the Cox-Voinov theory may provide an explanation.

I also wonder about the literature on the friction from droplets moving on liquid-infused surfaces were both a linear relationship between friction and speed (a classical Stokes-type law), and friction varying as a $2/3$ power law of speed has been discussed. I understand the current manuscript does not use liquid-infused surfaces and so the authors may regard this comment as outside the scope of the manuscript. However, some of the discussion about friction laws may be of interest. Relevant literature here includes a paper by one of the current co-authors, Keiser, A.; Baumli, P.; Vollmer, D.; Quéré, D. “Universality of Friction Laws on Liquid-Infused Materials”. *Phys. Rev. Fluids* 2020, 5 (1), 014005. and Smith, J. D.; Dhiman, R.; Anand, S.; Reza-Garduno, E.; Cohen, R. E.; McKinley, G. H.; Varanasi, K. K. “Droplet Mobility on Lubricant-Impregnated Surfaces”. *Soft Matter* 2013, 9 (6), 1772–1780.

Thanks for the comments. Eq. 4 on page 10 is the Furmidge-Kawasaki equation, $F_c = k\omega\gamma(\cos\theta_r - \cos\theta_a)$, a simplified formula from integrating unbalance force around the contact line. It is not immediately clear to us why eq.4 can almost cover all the kinetic friction force. Thus, it is instructed to discuss the possible reason though it is qualitative. Our explanation is that relevant dissipating processes act close to the contact line and influence the apparent contact angles, which enter into eq. 4. The relevant dissipations may include not only viscous dissipation discussed by Cox-Voinov theory but also contact-line friction described by molecular kinetic theory or their combination. Eq. R1 is obtained by a Taylor expansion of eq.4 and assuming dynamic contact angles follow Cox-Voinov theory. Essentially it only considers dynamic contact angles in eq. 4 contain the contribution of viscous dissipation. This has been included in our discussion on page 10. In addition, we have discussed the possibility of combining the Cox-Voinov theory and the Furmidge-Kawasaki equation in the revised main text. For details, please refer to our response to comment 2.

Regarding the discussion about friction laws in the above literature, a non-linear relationship between friction and velocity is founded due to the change in the shape of the lubricant meniscus for the liquid-infused surfaces. That is indeed out of the scope of this manuscript. However, to

broader our discussion about friction, we have added a discussion on page 13, which reads: “These above results indicate that viscous dissipation near the contact line is not the only dissipation channel changing the kinetic contact angles, at least not for water and low-viscous liquid. Only viscous dissipation is insufficient to account for drop friction. Other effects such as contact-line friction^{20, 48} or adaptation²³ contribute substantially. In addition, when the contribution from other effects increases substantially, the linear relationship between drop friction and velocity might change. For example, Keiser *et al.*⁴⁹ and Smith *et al.*⁵⁰ reported a nonlinear relationship between drop friction and velocity on liquid-infused surfaces due to the velocity-dependent change in the shape of the meniscus.”

REVIEWERS' COMMENTS

Reviewer #1 (Remarks to the Author):

The authors addressed all my concerns, I recommend the paper for publication in Nature Communications.

Reviewer #4 (Remarks to the Author):

The authors have made a sincere attempt to address my comments. The responses are reasonable and the manuscript and supplementary information has been amended accordingly.